# Real-time assembly of ribonucleoprotein complexes on nascent RNA transcripts

Olivier Duss [1,2], Galina A. Stepanyuk [1], Annette Grot[3], Seán E. O'Leary[2,4], Joseph D. Puglisi [2] & James R. Williamson [1]

Cellular protein-RNA complexes assemble on nascent transcripts, but methods to observe transcription and protein binding in real time and at physiological concentrations are not available. Here, we report a single-molecule approach based on zero-mode waveguides that simultaneously tracks transcription progress and the binding of ribosomal protein S15 to nascent RNA transcripts during early ribosome biogenesis. We observe stable binding of S15 to single RNAs immediately after transcription for the majority of the transcripts at 35 °C but for less than half at 20 °C. The remaining transcripts exhibit either rapid and transient binding or are unable to bind S15, likely due to RNA misfolding. Our work establishes the foundation for studying transcription and its coupled co-transcriptional processes, including RNA folding, ligand binding, and enzymatic activity such as in coupling of transcription to splicing, ribosome assembly or translation.

[1] Department of Integrative Structural and Computational Biology, Department of Chemistry, and The Skaggs Institute for Chemical Biology, The Scripps Research Institute, La Jolla, CA 92037, USA. [2] Department of Structural Biology, Stanford University School of Medicine, CA 94305 California, USA. [3] Department of Research and Development, Pacific Biosciences Inc, Menlo Park, CA 94025, USA. [4] Department of Biochemistry, University of California, Riverside, CA 92521, USA. Correspondence and requests for materials should be addressed to J.D.P. (email: puglisi@stanford.edu) or to J.R.W. (email: jrwill@scripps.edu)

Many cellular functions rely on the formation of large protein-RNA complexes (RNPs), which is often coupled to fundamental processes such as transcription or translation. The assembly of compositionally heterogeneous RNPs begins with the transcription of the RNA and can occur through multiple parallel pathways. Transcription kinetics can influence the RNA folding pathway, which in turn affects both co- and post-transcriptional assembly of proteins or other ligands on the nascent RNA. Therefore, approaches are needed to observe directly the coupling between RNA synthesis and assembly process representing the physiological context for RNA folding.

Delineating the coupling between assembly processes and biopolymer synthesis presents a significant experimental challenge. Single-molecule experiments are real-time approaches that allow simultaneous monitoring of multiple compositional and conformational parameters for complex systems with high temporal (ms) resolution[1–3]. They have been extended beyond binary ligand-macromolecule interactions to monitor enzyme movement during transcription, translation, and replication and at the same time providing compositional and conformational information on those macromolecular machines in real time[4–18]. While these approaches have provided unprecedented insight into the structure-activity relationship of specific multicomponent systems, they suffer from one or more drawbacks: the lack of high-throughput measurements to capture rare events, limitation to low nM concentrations of fluorescently-labeled macromolecules, and limits on the number of components or observables that can be simultaneously monitored. Critically, the ability to work at higher, physiological ligand concentrations (>100 nM) is needed to define the kinetics of complex multistep processes such as the competition between RNA folding and protein binding that occur during co-transcriptional ribosome assembly or splicing. Data from sufficient numbers of molecules are crucial for statistically-robust analysis of these complex mechanisms.

Zero-mode waveguide (ZMW) fluorescence microscope technology allows single-molecule real-time dynamics of complex biological systems to be delineated at physiological ligand concentrations for thousands of single biomolecules simultaneously through four spectral channels[19,20]. This technology has been exploited for DNA sequencing[21], for studying translation by the ribosome[15,16] and for other applications[22]. By allowing high-throughput single-molecule analysis at high ligand concentrations at high temporal (10 ms) resolution, processive reactions such as transcription and translation can occur efficiently, and different reaction pathways can be observed directly.

Here we have developed a general method to track both transcription and the simultaneous assembly of proteins on the nascent transcript using ZMW technology. Stalled transcription complexes, comprising a DNA template, RNA polymerase (RNAP), and a short leader transcript, were immobilized in ZMWs, to allow observation of protein binding to single nascent RNAs (Fig. 1a, b). Transcription was initiated by releasing the stalled complex with the addition of NTPs and the simultaneous addition of fluorescently-labeled ligands that can interact with the growing nascent transcript in real time (Fig. 1b). ZMWs are nanophotonic structures that generate a sharply decaying illumination profile in a direction normal to the surface. By labeling the DNA template at either the 5′- or 3′-ends, transcription can be monitored in real time by a fluorescence intensity change as the labeled DNA template moves through the evanescent field gradient (Fig. 1c). This system allows for considerable flexibility in monitoring specific steps during the reaction by the choice and location of the fluorescent dyes on the DNA, RNA, and protein components. Using this approach, we have developed assays that allow us to monitor simultaneously the progress and rate of transcription, the formation of full-length RNA transcripts,

timing of transcriptional pausing at terminators, release of the DNA template, and specific binding of proteins at >100 nM concentration and on hundreds to thousands of single rRNAs in parallel during a single experiment.

The complex process of ribosome assembly involves co-transcriptional binding of over fifty ribosomal proteins (r-protein) to the ribosomal RNA (rRNA). As a model system, we monitor here the binding of *E. coli* ribosomal protein S15 to nascent transcripts of the central domain of 16 S rRNA. Our results show that many transcripts do not bind to S15 at 20 °C. For other nascent RNA molecules, S15 binding occurs immediately after termination of transcription, however not co-transcriptionally. We find two distinct modes of S15 binding, which are rapid and transient association and dissociation of S15, and kinetically stable binding of S15. Overall, this approach provides an unprecedented view of the individual molecular events that occur during protein binding to a nascent RNA transcript, providing a powerful platform to understand the mechanism of co-transcriptional ribosome assembly, as well as other complex macromolecular processes.

## Results

**Monitoring transcription in an evanescent field**. Stalled transcription elongation complexes were prepared in solution using a DNA template that contained the *E. coli* P1 promoter at the 5′-end, and a transcription terminator at the 3′-end (Fig. 1a). Transcription by *E. coli* RNA polymerase (RNAP) was initiated with nucleotide triphosphates (NTPs), omitting GTP, resulting in a complex with RNAP stalled at the first guanine nucleotide. This complex bears a nascent transcript of 50 nucleotides and is sufficiently stable to be visualized on a native gel (Supplementary Fig. 1b). The transcriptional stall can be conveniently released by addition of a reaction mixture containing all four NTPs (Supplementary Fig. 1a, b), while transcription re-initiation is prevented by addition of rifampicin. The transcription terminators allow efficient dissociation of the DNA template and polymerase from the nascent RNA (Supplementary Fig. 1b). The resulting full-length transcript can be visualized by a fluorescently-labeled DNA oligonucleotide hybridized to the 5′- or 3′-end (Supplementary Fig. 1b).

For single-molecule experiments, we immobilized the stalled transcription complexes at the bottom of biotin-functionalized ZMWs coated with NeutrAvidin, either using a biotinylated DNA oligonucleotide hybridized to the 5′-end of the nascent RNA of the stalled complex or by priming transcription with a biotin-ACU trinucleotide (see "Methods" section). Real-time single-molecule experiments were initiated by delivering all four NTPs to the immobilized, stalled transcription elongation complex (Fig. 1b). To monitor the progression of transcription and dissociation of the DNA template, the distal 3′-end of the DNA template was labeled with Cy3.5. The DNA template was labeled with two Cy3.5 dyes to distinguish Cy3.5 photobleaching that results in a two-step decrease in Cy3.5 intensity from template dissociation that results in a single-step disappearance of Cy3.5 fluorescence.

Consistent with a transcription elongation-dependent movement of the template 3′-end into the excitation volume close to the ZMW surface (Fig. 1c), we observed a monotonic increase in the Cy3.5 signal (Fig. 1d, e and Supplementary Fig. 8), typically starting within a few seconds after NTP delivery (Supplementary Fig. 1g). The Cy3.5 fluorescence intensity increase during transcription is specific for transcription elongation as no such behavior was observed in absence of all 4 NTPs. Movement of the DNA template within the evanescent field in the ZMW can be monitored from either the 5′-end, the 3′-end, or both ends, using

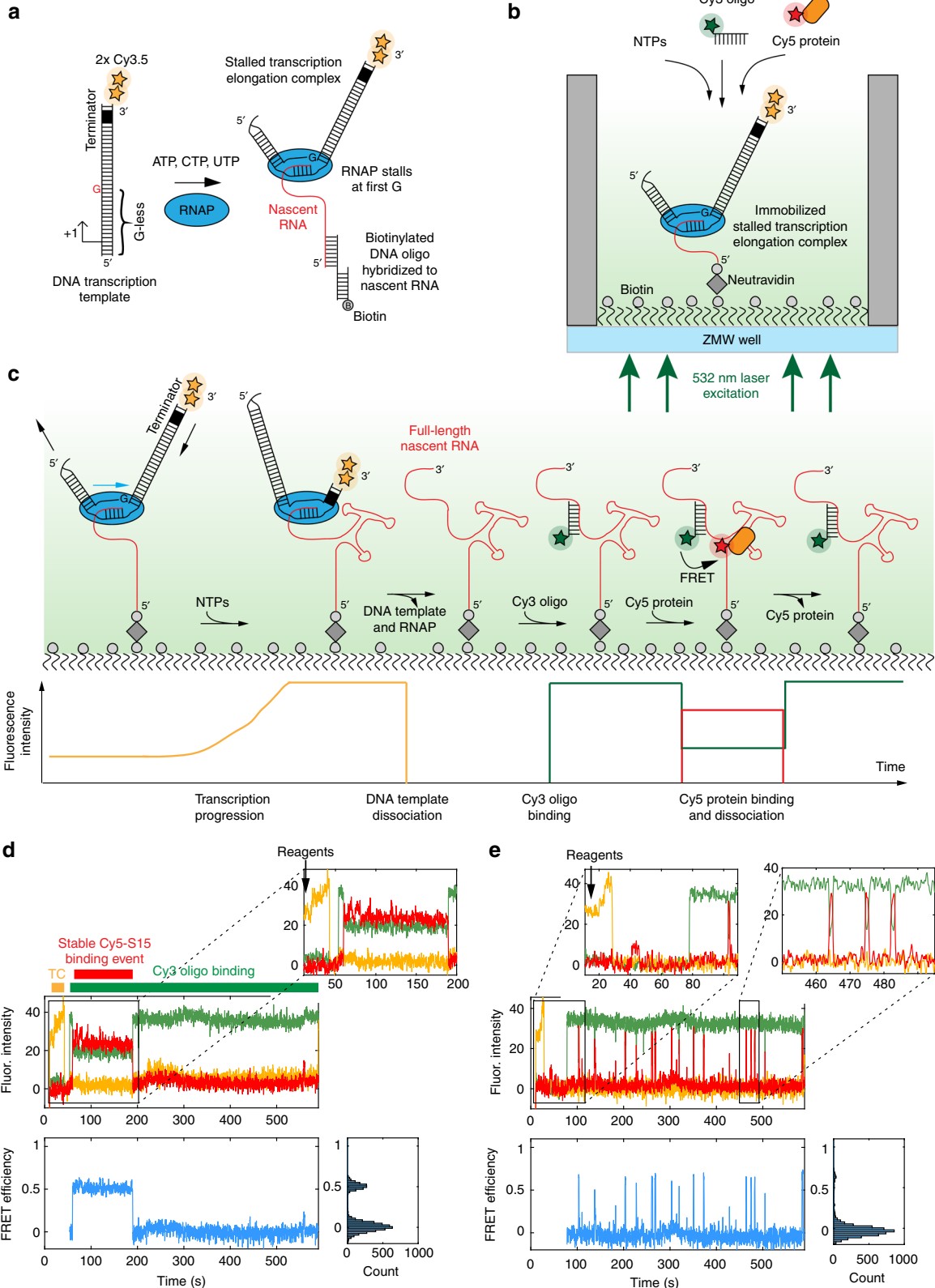

**Fig. 1** Experimental approach for monitoring transcription of and protein binding to single RNA molecules. **a** Stalled transcription elongation complex formation. **b**, **c** Experimental setup to observe transcription and protein binding to the same nascent RNA molecule in real time. **d**, **e** Representative single-molecule traces with fluorescence intensity (top), FRET efficiency (bottom) and its probability distribution (bottom right) for a single transcribing RNA molecule binding to 5 nM r-protein S15, representing a natively folded (N-class) (**d**) or partially folded (P-class) nascent RNA molecule (**e**). Transcription elongation (TC, orange), Cy5-S15 binding (red) and Cy3-oligo binding (green) are indicated as bars in (**d**). NTPs, Cy3-oligo and Cy5-S15 protein (reagents) were delivered simultaneously at 14.4 s (**b**), which is indicated by an arrow in (**d**, **e**). A single laser at 532 nm was used to directly excite both the Cy3 and Cy3.5 dyes

different combinations of dyes, producing corresponding fluorescence intensity increases or decreases depending on the labeling scheme (Fig. 2a–d).

The observed dye fluorescence intensity is strongly dependent on the distance from the surface of the ZMW (Supplementary Fig. 2a, b). This physical property allowed monitoring of transcription progression by a change in fluorophore position along the axis normal to the surface. We quantified the fluorescence intensity change during transcription, over the interval from NTP delivery to template dissociation, for seven transcripts ranging from 171 to 517 nucleotides in length (Fig. 2e). We found that for a population of equal-length transcripts, the intensity change associated with completed transcription varied widely across the chip (Fig. 2f), due to optical distortion in the instrument along with any residual misalignment of the chip to the illumination[23]. Consequently, the mean absolute fluorescence intensity change during transcription was poorly correlated with transcript length (Supplementary Fig. 2f). To account for the varying illumination intensities across the chip, we normalized the intensity change during transcription by the fluorescence intensity change during template dissociation for each individual molecule. The relative intensity change during transcription accounted for the effects of variable illumination power, and resulted in a tightly distributed value for RNAs of the same length (Fig. 2g, Supplementary Fig. 2 and Supplementary Note 1). Thus, plotting the normalized intensity change during transcription against transcript length provides a "calibration curve" (Fig. 2h). This curve can be well fitted to a model in which the dyes at the 3'-end of the DNA template move within the evanescent field closer to the surface (Supplementary Note 2 and Supplementary Fig. 2). Using a semi-empirical model, which treats the DNA as a worm-like chain (WLC)[24], we obtain an evanescent decay constant $c$, which is in agreement with calculations of the decay constant within the ZMW holes using a finite-difference time-domain approximation to Maxwell's electromagnetic equations (Supplementary Figs. 2c,d and Supplementary Notes 1 and 2). Overall, the normalized intensity change in the ZMWs during transcription elongation is a defined quantity that reports on the number of nucleotides transcribed.

To mark the transcription end-point independently, a Cy3-labeled DNA oligonucleotide targeting the 3'-end of the full-length RNA was simultaneously added with the NTPs at the beginning of the experiment (Fig. 1b, c). Its fast bimolecular association rate ($5.2 \pm 0.1 \times 10^5\,\mathrm{M}^{-1}\,\mathrm{s}^{-1}$; the error representing the 95% error bounds of the fit) ensured rapid detection of full-length RNA (Supplementary Note 3). Combining two signals monitoring both transcription progress and production of full-length RNA, these results demonstrate our ability to observe transcription in real time.

**Modulating transcription rates**. We next investigated transcription elongation rates as a function of transcript length, NTP concentration, and temperature. The dwell times during transcription progression for different molecules fit well to a normal distribution, with the mean representing the average transcription time (Fig. 3a and Supplementary Note 4). For example, transcription of a 517 nucleotide construct (see Supplementary Methods) at 0.5 mM NTPs requires on average $20.1 \pm 4.3\,\mathrm{s}$, implying an average transcription rate of 25.6 nt s⁻¹ at 20 °C (Fig. 3a). We measured the mean dwell times for transcription of DNA templates with different lengths, spanning 72-517 base-pairs, revealing a near-linear correlation (Fig. 3b, Supplementary Note 5). These data are consistent with the absence of long >10 s sequence-specific transcription pauses and a mean transcription rate that is largely independent of the length and sequence of the template at saturating NTP concentrations for construct lengths between 100 and 500 nt.

To determine the NTP-dependence of transcription, transcription rates for a 171 nt RNA were measured at NTP concentrations ranging from 25–1000 μM. The mean rates varied between 0.4–16.3 nt s⁻¹ for this construct (Fig. 3c). The rate as a function of the NTP concentration reaches apparent saturation at high nucleotide concentrations, with values comparable to previous bulk in-vitro transcription and single-molecule studies (10–36 nt s⁻¹ ref.[11,25–28]).

Next, we investigated the temperature dependence of the transcription rate using the 517 nt construct. The rate increased from an average of 25.6 nt s⁻¹ to 56.2 nt s⁻¹ upon increasing the temperature from 20 °C to 35 °C (Fig. 3d). We fit this temperature dependence to the Arrhenius equation and obtained an activation energy for transcription elongation of $9.6 \pm 1.7\,\mathrm{kcal\,mol}^{-1}$. This is in good agreement with a previous single-molecule study using optical tweezers reporting $9.7 \pm 0.7\,\mathrm{kcal\,mol}^{-1}$ (ref.[29]), and with the reported activation energy for the rate of RNA elongation in vivo of $\sim10.9\,\mathrm{kcal\,mol}^{-1}$(ref.[27]).

These results show that we can monitor real-time transcription of single RNAs from DNA templates ranging from ~100 to 500 nucleotides, at transcription rates spanning more than two orders of magnitude (0.4–60 nt s⁻¹), by controlling NTP concentration or temperature.

**RNA polymerase stalling at transcription terminators**. Completion of transcription elongation was typically followed by a sudden drop in Cy3.5 fluorescence intensity (Fig. 4a) due to dissociation of the Cy3.5-labeled DNA template and therefore dissociation of the RNAP from the nascent RNA (Supplementary Fig. 1b). However, a subset of molecules exhibited a prolonged plateau of fluorescence between the end of transcription and template dissociation (Fig. 4b). Using our standard construct containing a triple terminator (*M13central*++, *rrnBT1* and *T7Te*) followed by two Cy3 dyes at the 3' terminus (Supplementary Fig. 3), we found approximately 5–15% of the traces showing this intensity plateau, which persisted for several minutes, and in some cases even to the end of the 10-min movie. This indicates prolonged stalling of the RNAP at the 3'-end before dissociation. For the remaining molecules, RNAP dissociation occurred within $3.2 \pm 0.8\,\mathrm{s}$ on average after the end of transcription (Fig. 4c). We could modulate the RNAP dissociation kinetics by changing the structural context at the terminator increasing the RNAP residence time at the terminator to $20 \pm 5\,\mathrm{s}$ (Fig. 4c, Supplementary Fig. 3 and Supplementary Note 6). The fast dissociation kinetics observed at the triple terminator are in agreement with a previous single-molecule study showing that no terminal dwell was found for the *his* or *tR2* terminators and a delay of 0-4 s was measured for the *t500* terminator depending on the applied force[30]. Our results demonstrate that we can separately observe transcription progression and dissociation of the RNA polymerase during termination of transcription.

**S15 binding to pre-folded RNA**. To compare protein binding to a pre-transcribed and pre-folded RNA with binding to a nascent RNA emerging from the RNAP, we investigated the interaction of the r-protein S15 with the 3-helix junction composed of helices 20, 21, and 22 in the central domain of the *E. coli* 16 S rRNA (Fig. 5a, b). S15 is a r-protein that nucleates assembly of the central domain by organizing the conformation of the 3-helix junction[31,32], allowing subsequent assembly of other r-proteins[33]. The S15•3-helix junction complex represents an excellent well-characterized biochemical system, with a significant body of data on binding thermodynamics and specificity[34–37]. Previous single-

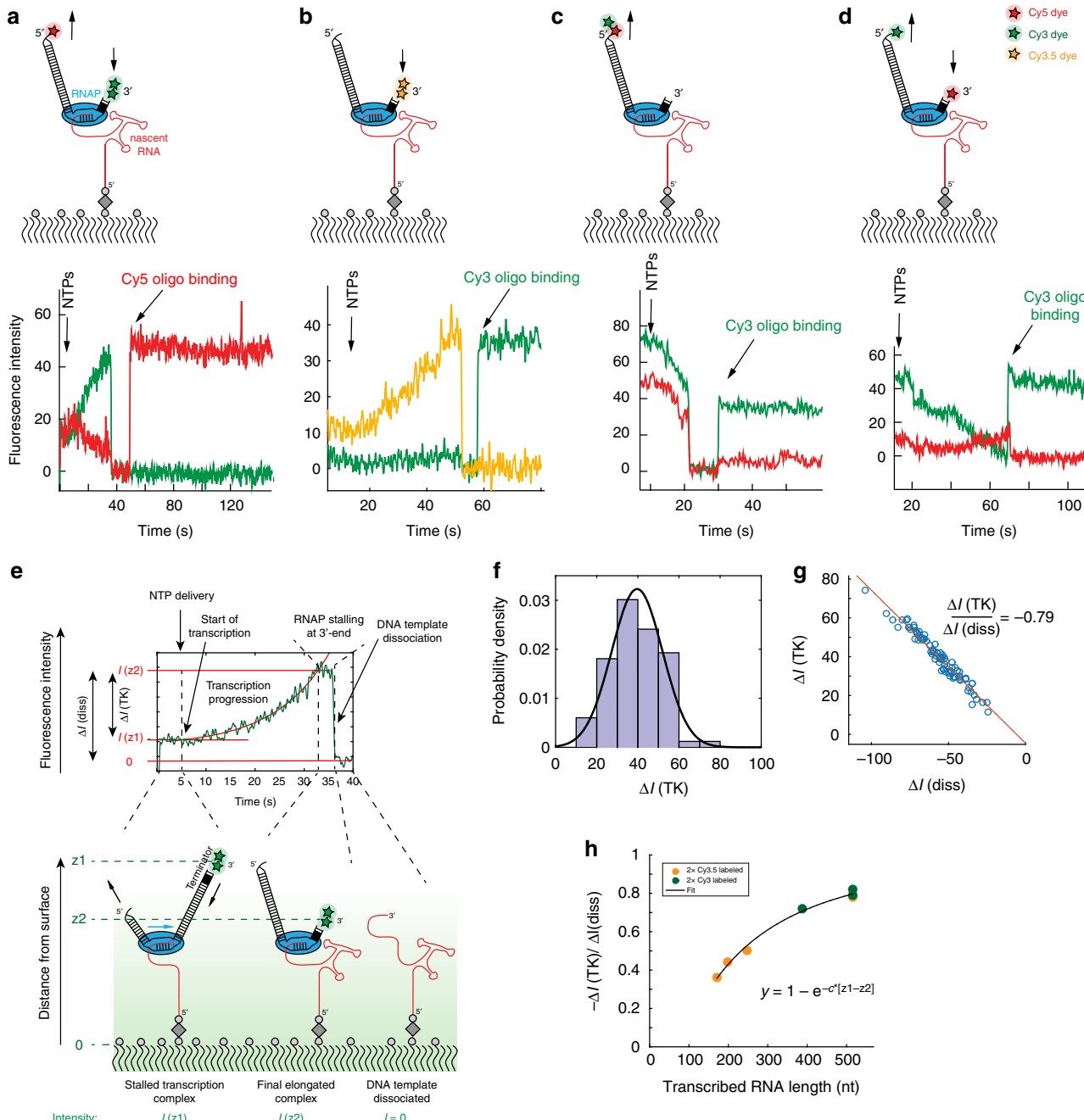

**Fig. 2** Transcribing within the evanescent field. **a–d** Different DNA template labeling schemes result in different fluorescence intensity profiles. The DNA template labeling schemes are shown at the top of each trace with arrows showing the movement of the corresponding dyes towards or away from the surface during transcription progression resulting into a fluorescence intensity increase or decrease, respectively. The times of NTP injection and of labeled DNA oligo binding are indicated with arrows. The DNA oligo binds to the 3′-end of the nascent RNA and thereby scores for full-length RNA as illustrated in Fig. 1. Note that the characteristic fluorescence intensity changes also allow using a single color for monitoring transcription progression and scoring of full-length RNA with the labeled DNA oligo, i.e. Cy3 in (**c**, **d**). 532 nm and 642 nm lasers were used to excite the Cy3 and Cy5 dyes, respectively (**a**, **c**, **d**). A single laser at 532 nm was used to excite both the Cy3 and Cy3.5 dyes (**b**). **e** The total fluorescence intensity change during transcription, ΔI(TK), is related to the dye position change along the evanescent field. ΔI(diss) denotes the intensity change upon DNA template dissociation. **f** The absolute fluorescence intensity change during transcription widely varies between transcribed RNA molecules of the same size due to inhomogeneous illumination intensity across the ZMW chip. **g** Normalizing the intensity change during transcription by the intensity change during DNA template dissociation results in a nearly constant value (slope of curve) for all the molecules of the same length. **h** The normalized intensity change during transcription relates to the transcribed RNA length by a model in which the DNA template moves within the evanescent field (see Supplementary Fig. 2 and Supplementary Note 2). For (**f**, **g**), a 517 nt RNA was transcribed from a DNA template labeled as shown in (**e**); Number of molecules analyzed (*n*) = 83 (**f**, **g**) and (*n*) = 109, 65, 78, 72, 53, 70, 83 for the data points at 171, 199, 247, 388, 517, 517 and 517 nts transcribed RNA lengths (**h**). Source data are provided as a Source Data file

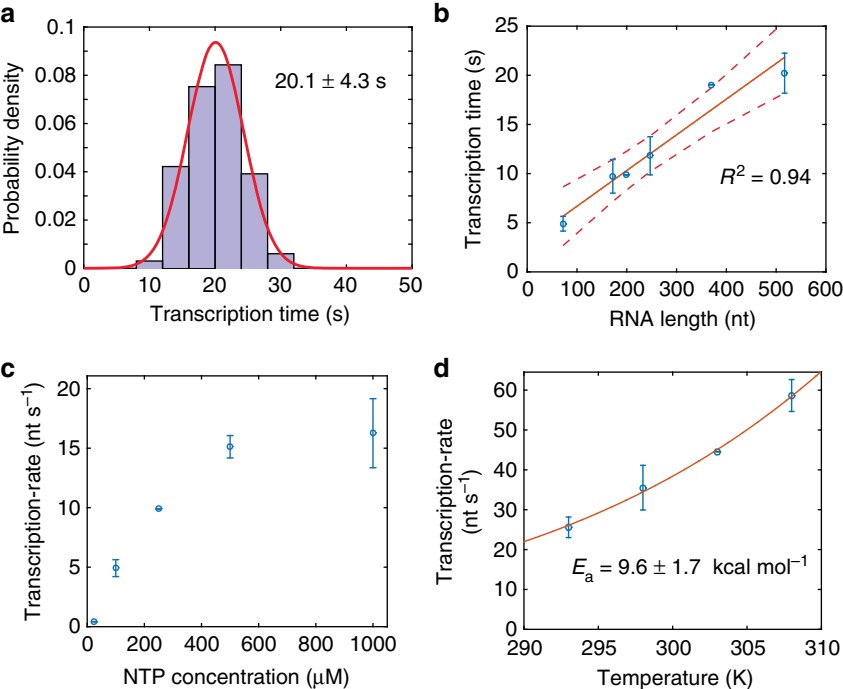

**Fig. 3** Transcription progression. **a** The time to transcribe an RNA of a specific length, here the 517 nt RNA construct, is broadly distributed. The mean and standard deviation of the Gaussian distribution are shown. Number of molecules analyzed ($n$) = 83. **b** The average time to transcribe an RNA is proportional to its length. 95 % prediction intervals of the fit are shown. **c** Effect of NTP concentration on transcription rate for the 171 nt RNA construct. **d** Effect of temperature on transcription rate shown for the 517 nt RNA construct. The data were fitted to the Arrhenius equation. For all the plots, error bars represent the standard deviation obtained from biological replicates. Source data are provided as a Source Data file

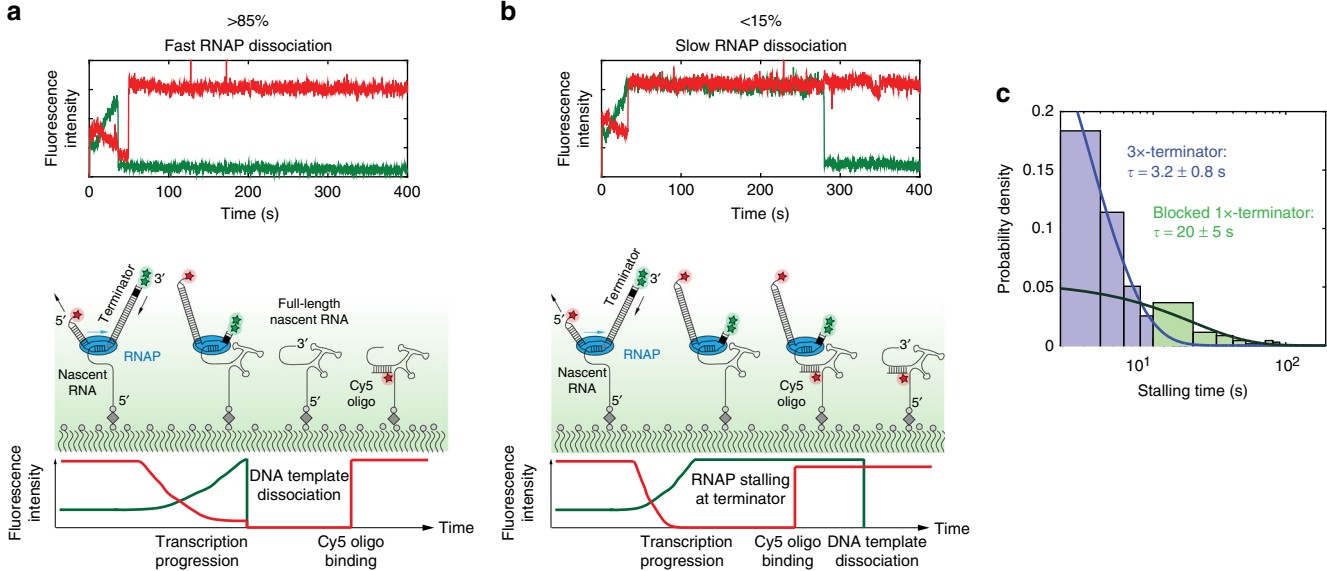

**Fig. 4** RNA polymerase stalling. **a, b** Representative single-molecule traces (top) and corresponding molecular events (bottom) for fast (**a**) and slow (**b**) RNAP dissociation. 532 nm and 642 nm lasers were used to excite the Cy3 (green) and Cy5 (red) dyes, respectively. **c** Residence times of the polymerase at the 3′-end of the DNA template before dissociation depends on the identity of the 3′-end. Only residence times < 100 s were used for fitting to a single exponential function and the stalling time is shown in log-scale. Number of molecules analyzed ($n$) = 79 and 71 for 3 × -terminator and blocked 1 × - terminator, respectively. Constructs and histograms in linear scale are shown in Supplementary Fig. 3. Source data are provided as a Source Data file

molecule studies have investigated the influence of $Mg^{2+}$ ions or S15 on the conformational dynamics of the 3-helix junction, monitoring fluorescence resonance energy transfer (FRET) changes between different helical arms[38–41]. However, the binding kinetics of single S15 r-proteins have not yet been observed. More importantly, previous experiments for S15 binding were

performed on a pre-transcribed and pre-folded rRNA and not a nascent RNA transcript, as would be the case during co-transcriptional ribosome assembly in a living cell.

We first investigated the binding of S15 to a pre-folded in-vitro transcribed RNA (171 nt construct; see Methods). The S15(T79C) variant was labeled with Cy5-maleimide, retaining full RNA

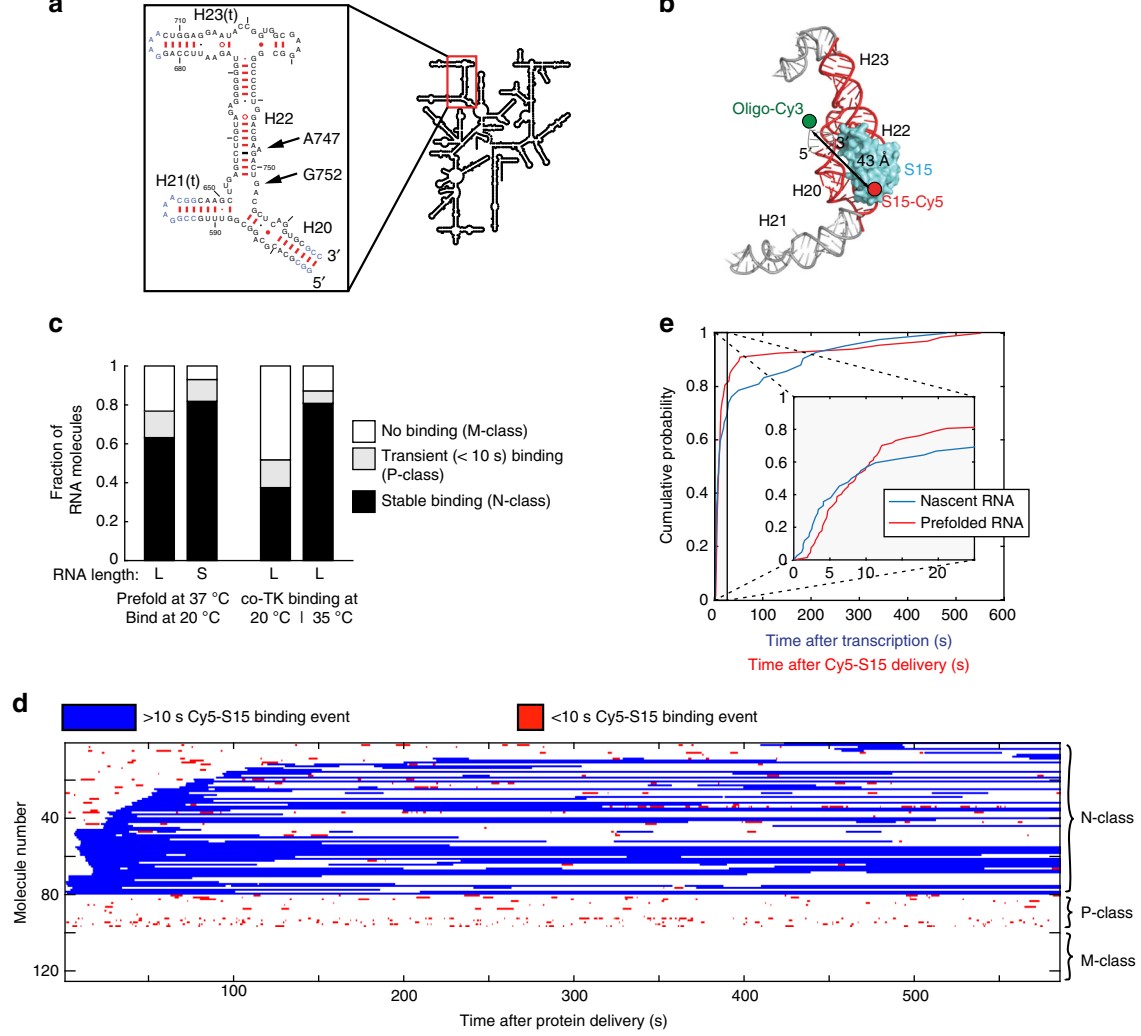

**Fig. 5** Comparing S15 binding to pre-folded versus nascent RNA. **a** RNA construct used to observe S15 binding: It constitutes part of the central domain of the 16 S rRNA. H21(t) and H23(t) denote truncated helices compared to wild-type. RNA mutants used in this study are depicted. **b** Cy5-labeled r-protein S15 bound to the 16 S rRNA of the *E. coli* 30 S ribosomal subunit (PDB accession code: 4V9P); the Cy5 dye of the protein and the Cy3 dye of the labeled DNA oligonucleotide bound to the 3′-end of the nascent RNA are modeled into the structure. **c** Partitioning of the RNA molecules into three folding classes for pre-folded compared to nascently transcribed RNA. N-class: natively folded RNA; P-class: partially folded; M-class: misfolded. L is 171 nt and S denotes 85 nt construct (see Supplementary Methods). Number of molecules analyzed (n) = 125, 65, 168, 172 from left to right. **d** Timing of S15 binding events in a stack of single-molecule traces, where 5 nM Cy5-S15 protein was delivered at t = 0 s to a pre-folded, immobilized RNA. Traces were clustered by lifetime and initial onset of stable (>10 s) S15 binding events and the total number of events per trace, resulting in three RNA folding classes. Each row represents the trace resulting from a single pre-folded RNA molecule. **e** Time of first stable S15 binding event after delivery of 25 nM Cy5-S15 protein to a pre-folded RNA (red) or after transcription of a nascent RNA (blue), respectively. For nascent RNA binding, the experimental scheme in Fig. 6a was used. Number of molecules analyzed (n) = 67 and 42 for pre-folded or nascent RNA, respectively. Source data are provided as a Source Data file

binding activity[42,43]. We hybridized a Cy3-labeled DNA oligonucleotide to the 3′-end of the RNA, which allowed S15 binding to be monitored by FRET between the Cy3-DNA and Cy5-S15. The distance between residue 79 on S15 and the 3′-end of helix 20 in the three-helix junction RNA construct is ~43 Å based on the structure of the intact 30 S ribosomal subunit (Fig. 5b). Importantly, the application of a FRET signal permitted specific S15-RNA binding to be unambiguously distinguished from nonspecific interaction of labeled S15 with the ZMW surface.

At 20 °C, ~65% of the pre-folded RNA molecules bind S15 stably (denoted N-class for "Natively folded" RNA molecules), while other molecules show bursts of transient S15 binding (~15%, P-class for Partially folded) or no protein binding at all (~20%, M-class for Mis-folded) (Fig. 5c, d and Supplementary Fig. 4). We define kinetically stable binding by an S15-bound

lifetime of >10 s (see "Methods" section). Using a shorter 85 nt RNA construct[40,43] (Supplementary Methods and Supplementary Fig. 5a), we found that > 80 % of the RNA molecules stably bind S15 (Fig. 5c), which is consistent with a smaller probability of RNA misfolding for a shorter RNA.

The N-class transcripts exhibit few long-lived (>10 s) S15 binding events, along with rare short events of a few seconds (Fig. 5d). Of those molecules, 70–90 % exhibit a final S15 binding event, the lifetime of which is either limited by the end of the 10-minute movie or by Cy5 photobleaching (Fig. 5d, Supplementary Note 7). The distribution of S15 binding-event durations suggested an upper limit for $k_{off} = 6.0 \pm 0.7 \times 10^{-3}$ s$^{-1}$, corresponding to an average S15-bound lifetime of around 170 s (Supplementary Fig. 6a,b), in agreement with previous ensemble and single-molecule experiments[32,42] (Supplementary Note 8). At

25 nM Cy5-S15, the first stable S15 binding occurs on average less than 10 s after delivery of the protein to the immobilized RNA (Fig. 5e). By fitting the time between Cy5-S15 protein delivery and the appearance of the first stable binding event to an exponential function, we can estimate an average S15 binding on-rate of $5.4 \pm 0.3 \times 10^6 \, M^{-1} \, s^{-1}$ (Supplementary Fig. 7a and Supplementary Note 8), which is in a similar range to previous ensemble experiments[32,42] (Supplementary Note 8).

Binding of S15 to a natively folded RNA includes proper folding of the three-way junction between helices 20, 21, and 22[31,36,37]. Using real-time monitoring of the RNA conformational changes upon S15 binding, we show that the RNA is in the closed native conformation during the stable binding events (Supplementary Fig. 5a-c). Furthermore, in a G752C RNA mutant that is unable to form the 3-way junction (Fig. 5a)[36,44], S15 protein binding was completely abolished in our single-molecule experiments. We therefore conclude that RNA molecules that stably bind S15, representing the majority of the pre-folded RNAs, are natively folded.

**Correlating transcription and protein binding to nascent RNA.** Next, we investigated binding of Cy5-S15 to an RNA that is being actively transcribed in the presence of the S15 protein. After immobilization of a stalled transcription complex, we started the experiment by addition of 0.5 mM NTPs and various concentrations of Cy5-S15, its specific binding again detected with a FRET to a simultaneously added Cy3-labeled DNA oligonucleo-tide binding to the 3′-end of the nascent RNA (Fig. 1). Concomitantly with (Supplementary Fig. 11e), or shortly after Cy3 oligo binding (Fig. 1d, e), we start to detect Cy5-S15 binding events as anti-correlated Cy3 and Cy5 intensity changes, which can be followed for at least 10 min with almost no traces showing Cy3 oligo photobleaching.

In contrast to the pre-transcribed and pre-folded RNA, only ~40% of the nascent RNA molecules show stable S15 binding at 20 °C (N-class; Fig. 5c, Supplementary Fig. 4, Supplementary Fig. 6g). For those molecules, we obtained a similar average S15-bound lifetime to that seen for the in-vitro transcribed and pre-folded RNA (Supplementary Fig. 6a,b). This suggests that this population of nascent RNA molecules is natively folded (N-class). To demonstrate further that stable S15 binding events are due to binding to a natively folded RNA, we repeated the experiment with an RNA variant with A747 deleted (Fig. 5a), which is a linker nucleotide within H22 required for S15 to bind with high affinity to both the lower and upper part of H22[36,44] and has a 2.5-fold lower affinity compared to wild-type[44]. This mutation resulted in 5-fold faster S15 dissociation with $k_{off} = 2.8 \pm 0.1 \times 10^{-2} \, s^{-1}$, or an S15-bound lifetime of around 36 s for the N-class RNA molecules but did not affect the relative abundance and kinetic binding behavior of the other co-transcriptional folding classes (P- and M-class), (Supplementary Fig. 4 and Supplementary Fig. 6a,b,g).

Upon Cy5-S15 binding, we observe an immediate transition into a high-FRET state with apparent FRET efficiencies of approximately 0.5–0.6. This is a bit lower than expected from the distance between the Cy3 and Cy5 dyes in the native complex but is in agreement with a decreased FRET value observed in the ZMW compared to TIRF due to quenching of the Cy5 fluorescence by the ZMW aluminum walls[15,20]. The primary binding protein S4 has been shown to bind first in a low-FRET state before switching to the native high-FRET state within ~1 s[3]. There is no evidence for such an initial binding in a low-FRET state indirectly observing Cy5-S15 by FRET from the Cy3-oligo. To further address possible binding through an intermediate low-FRET state, we repeated our experiments using direct excitation

of both Cy3 and Cy5 dyes at low 2 nM Cy5-S15 concentration (Supplementary Fig. 12a). We see a direct binding into the high-FRET state and no initially bound S15 protein without FRET in > 95 % of the traces. Overall, those experiments support the binding of S15 directly into a high-FRET state, corresponding to the native RNA conformation.

We investigated how long after completion of its synthesis a specific nascent RNA molecule requires to obtain a S15-binding competent fold. We found that stable S15 binding occurs shortly after transcription (Supplementary Fig. 4); notably, its detection is often limited by the binding of the Cy3 DNA oligonucleotides to the 3′-end of the nascent RNA (Supplementary Fig. 7c). In order to detect protein binding specifically during transcription, we modified our experimental setup by hybridizing a Cy3 DNA oligonucleotide to the 5'-end of the nascent RNA in the stalled transcription complex (Fig. 6a, Supplementary Fig. 13 and Methods). Then, we initiated transcription with the addition of all NTPs and 25 nM Cy5-S15 protein. We did not detect any specific Cy5-S15 binding during transcription, evidenced by the absence of FRET between the Cy3 DNA oligonucleotide and Cy5-S15 in > 99 % of the traces. However, we observed kinetically stable S15 binding a few seconds following transcription completion (Fig. 5e and Fig. 6b, c). Remarkably, the S15 arrival time distribution is very similar to S15 binding to a pre-folded RNA (Fig. 5e): We obtain a second order binding rate constant with a main phase of $6.0 \pm 0.4 \times 10^6 \, M^{-1} \, s^{-1}$ (68%) by fitting the distribution of arrival times from end of transcription until the first stable binding event (Supplementary Fig. 7b and Supplementary Note 8). Those findings demonstrate that the N-class molecules of the 3-way junction central domain RNA fold into their native conformation almost immediately after transcription.

Based on the S15 binding behavior, co-transcriptional folding of nascent RNA is biased toward the P- and M-classes. In addition to the 40 % natively folded RNA molecules (N-class) obtained co-transcriptionally at 20 °C (Fig. 1d), approximately 40–50 % of the nascent RNA molecules were completely unable to bind S15 (M-class) and another ~10–20 % of nascent RNA molecules (P-class) have very distinct S15 binding kinetics compared to the natively folded molecules (Fig. 1e and Supplementary Fig. 6d), exhibiting bursts of short-lived binding events (Fig. 1e and Supplementary Fig. 4) with an average lifetime of 1–1.5 s (Supplementary Fig. 6c), but devoid of stable binding. The Cy5-S15 binding rate for the P-class RNA molecules depends linearly on the Cy5-S15 concentration, with an apparent second order binding rate constant of $4.8 \pm 0.4 \times 10^6 \, M^{-1} \, s^{-1}$ (Supplementary Fig. 6e, f). This concentration dependence is in agreement with the fluctuations between zero-FRET and high-FRET corresponding to rebinding events rather than conformational changes during a long S15 binding event. To confirm the presence of multiple rebinding events, we have repeated the experiments with direct excitation of both Cy3 and Cy5 dyes at low 2 nM Cy5-S15 concentration (Supplementary Fig. 12b). No Cy5 fluorescence was detected between the short high-FRET state binding events, supporting the interpretation of multiple rebinding events.

Those RNA molecules with bursts of transient S15 binding have at least a partially formed 3-way junction because S15 binding is entirely abolished in a G752C 3-way junction mutant and we do not detect transient S15 binding in that mutant. To obtain more direct structural information on the misfolding of those RNA molecules, we labeled H20 and H22 in the 3'-way junction construct with a Cy3 and Cy5 dye, respectively (Supplementary Fig. 5a,b). Ensemble and single-molecule experiments have demonstrated that S15 binding leads to closure of the H20/H22 helix conformation, which was manifested by a high-FRET state due to close proximity of the Cy3 and Cy5 dyes[38–40].

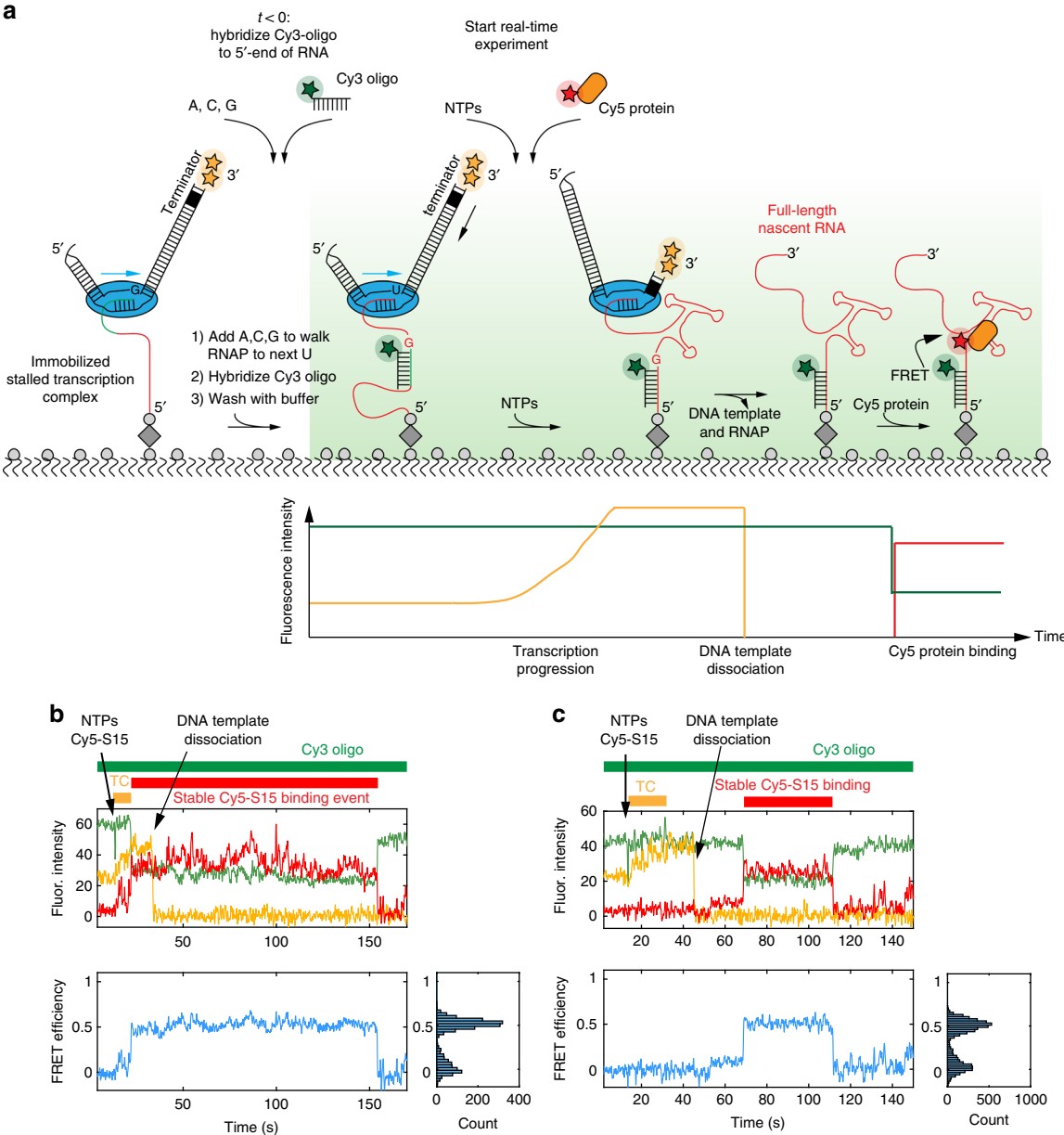

**Fig. 6** Experiment to detect true co-transcriptional binding. **a** Experimental setup for detection of specific Cy5-S15 protein binding during transcription (see Methods for more details). **b**, **c** Representative single-molecule traces with fluorescence intensity (top), FRET efficiency (bottom) and its probability distribution (bottom right) for a single transcribing RNA molecule binding to 25 nM r-protein S15 in which Cy5-S15 binds before (**b**) or after (**c**) DNA template dissociation. The periods of transcription elongation (TC, orange), Cy5-S15 binding (red) and Cy3-oligo binding (green) are indicated as bars. The reagents were delivered at 12.3 s. A single laser at 532 nm was used to directly excite both Cy3 and Cy3.5 dyes

Using this labeling scheme, we observed two types of FRET dynamics. The first type of trace, which represents the majority (>80%) of the RNA molecules, showed a transition from a low-FRET state to a high-FRET state shortly after S15 protein delivery and that persisted for several minutes (Supplementary Fig. 5c). This behavior corresponds to the kinetic signature we found for the N-class molecules that showed kinetically stable S15 binding shortly after protein delivery. In addition, we found ~10% of traces showing multiple transitions into the high-FRET state that persisted for only 1–2 s representing the kinetic binding behavior of S15 in the P-class RNA molecules (Supplementary Fig. 5d). Those data demonstrate that the RNA transitions into a native docked conformation, both during the stable S15 binding events in the N-class molecules (Supplementary Fig. 5c), and during the transient binding events in the P-class molecules (Supplementary

Fig. 5d). This multistep binding mechanism is consistent with an on-rate for S15 binding that is slower than diffusion controlled, in both N-type (Supplementary Fig. 7a, b) and P-type (Supplementary Fig. 6e, f) RNA molecules.

To investigate if the high fraction of RNA molecules unable to bind S15 is due to kinetic RNA folding traps at low temperature, we repeated our co-transcriptional binding experiment at 35 °C (Supplementary Fig. 4). The fraction of nascent RNA molecules binding S15 stably increased 2-fold to 80 % (Fig. 5c). As expected for protein-RNA dissociation, the S15 dissociation rate at 35 °C increased 5-fold with $k_{off} = 3.4 \pm 0.1 \times 10^{-2}\,s^{-1}$, or a S15-bound lifetime of ~ 30 s (Supplementary Fig. 6a, b). The remaining nascent RNA molecules are either S15 binding-incompetent (~15%, M-class) or show transient S15 binding (~5%). Our results demonstrate that the inefficient co-transcriptional RNA

folding at lower temperature is due to kinetic RNA folding traps (M-class), which can largely be overcome by increased thermal energy.

## Discussion

We present here a highly-multiplexed approach for monitoring transcription of thousands of single RNA molecules in real time during one experiment (Supplementary Fig. 1h, i), and the simultaneous binding of fluorescently-labeled proteins at near-physiological (>100 nM) concentrations. We measured transcription progression of RNAs ranging from < 100 to > 500 nucleotides at transcription rates varying over two orders of magnitude, and observed stalling of the RNA polymerase at the transcription terminator sequence. Long RNAs of several thousand nucleotides, such as full-length ribosomal RNAs, should also be accessible with our approach (Supplementary Note 5). Since this method does not require labeling or biotinylation of the RNA polymerase, it is also readily adaptable to the study of other RNA polymerases.

Our approach directly correlates transcription progression with protein binding to a nascent RNA. The progress of transcription is observed in real time by the movement of the labeled DNA template with respect to the ZMW surface, with resolution of tens of nucleotides, compared to single-nucleotide resolution in optical trap experiments[45]. RNAs significantly shorter than 100 nucleotides, will generate only a small intensity change during transcription progression, which may become difficult to detect. Yet, complementary to our approach, detection of transcription progression of very small RNAs (<30 nucleotides) in real time is

possible by following a FRET efficiency change between a 3'-end-labeled DNA transcription template and a labeled RNAP[13,18]. We readily observed S15 binding to RNA in the ZMW by monitoring a FRET signal between the Cy5-labeled protein and a Cy3-labeled DNA oligonucleotide that hybridizes to the transcript 3'-end or 5'-end. Since some nonspecific association of S15 to the surface of the zero-mode waveguides occurs, the FRET assay allows specific binding to be monitored even in the presence of a nonspecific background signal. In absence of non-specific protein binding, transcription and the binding of up to four different[20] labeled proteins to the nascent RNA could be monitored in the ZMW with the current setup by directly detecting protein binding[16,46,47] (Supplementary Fig. 12). This opens an avenue to delineate complex multistep protein-RNA assembly processes on nascent RNA, mimicking the co-transcriptional in-vivo situation.

Using this approach, we investigated the co-transcriptional binding kinetics of r-protein S15 to the 3-way junction between helices 20, 21 and 22 of 16 S rRNA (Fig. 7). Stable S15 binding occurs almost immediately after transcription, but we do not detect binding during transcription. This is not unexpected, as helix 20 consists of the 5'- and 3'-ends of the 3-way junction such that the complete 3-way junction required for S15 binding can only form after complete transcription (Fig. 5a).

High affinity binding by S15 requires proper folding of the three-way junction between helices 20, 21, and 22, a single-nucleotide bulge, and an internal loop in helix 22, but the non-Watson–Crick base-pairing in this region is not stably formed in the absence of S15[31,36,37]. It is possible that these non-native regions can form alternative interactions with other nucleotides in a manner that is incompatible with S15 binding. For example,

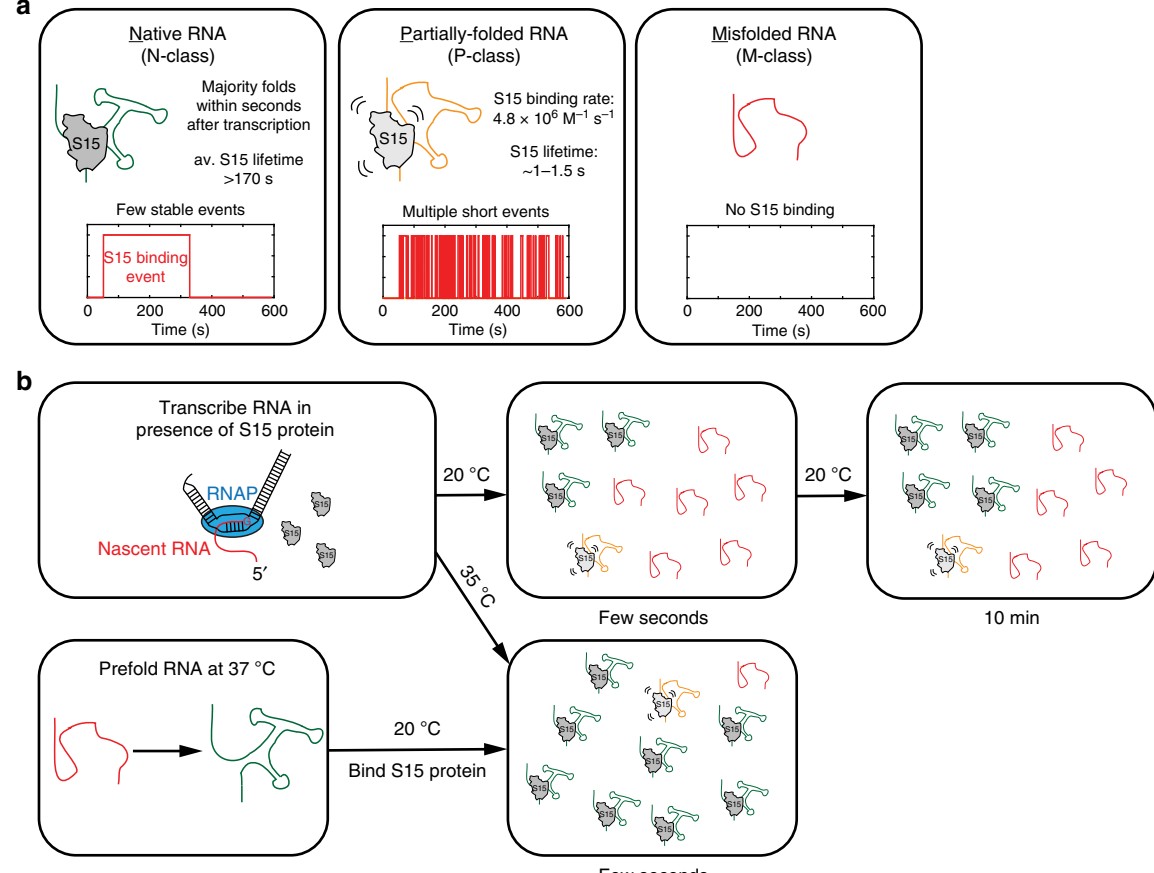

**Fig. 7** Model for S15 binding to nascent versus pre-folded RNA. **a** Kinetic characterization of different RNA folding classes. **b** Schematic of S15 binding to nascent versus pre-folded 3-way junction RNA

those alternative interactions could prevent proper formation of the 3-way junction: in a G752C RNA mutant that is unable to form the 3-way junction (Fig. 5a)[36,44], S15 protein binding was completely abolished in our single-molecule experiments. At 20 °C, more than half of the RNA molecules cannot form kinetically stable complexes with the 3-way junction binding-site but are either entirely binding-incompetent or show bursts of transient S15 binding. The majority of those misfolded RNA conformations disappear by increasing the temperature from 20 °C to 35 °C or pre-folding the RNA, suggesting that the kinetic barriers between native and misfolded RNA conformations are small and that RNA misfolding in this system is not based on long-lived non-native secondary structures[48,49].

Low-temperature stabilization is a commonly observed feature of misfolded, kinetically trapped RNAs[50–52], and our results demonstrate that kinetic traps can still form in nascently transcribed RNAs. Co-transcriptional folding has often been suggested as the mechanism by which cells can avoid misfolding that is frequently observed for RNAs prepared under denaturing conditions, in vitro[53–56]. Our results demonstrate that co-transcriptionally folded RNAs have distinct properties from refolded RNAs, but that some misfolding persists, at least in the case of the minimal S15 binding system. For the case of ribosome assembly, in vitro assembly rates[57,58] are >10-fold slower than the assembly reaction in cells[59]. Understanding how the co-transcriptional nature and participation of over 30 assembly cofactors guide ribosome assembly will be facilitated by our single-molecule co-transcriptional method. This approach can be extended to observe, in real time, full ribosome assembly, and other coupled transcriptional processes such as assembly of other ribonucleoprotein complexes and translation.

## Methods

**DNA template generation and labeling**. All the constructs are presented in Supplementary Methods. The DNA transcription templates were labeled at the 5'- and/or 3'-ends by hybridizing fluorescently-labeled DNA oligonucleotides to a single-stranded overhang on the double-stranded DNA transcription template. The labeled DNA oligonucleotides obtained from IDT were used without additional purification. The DNA template containing a single-stranded overhang was generated by autosticky PCR using a gBlocks gene fragment template available from IDT[60]. The PCR primers contained a sequence complementary to the DNA template, an abasic site, and the sequence complementary to the labeled DNA oligonucleotide. PCR was performed with Phusion DNA polymerase according to the manufacturer's instructions, using 35 cycles and optimizing the annealing temperature with a temperature gradient. The PCR product was purified on an agarose gel with subsequent QIAGEN gel extraction, followed by buffer exchange (10 mM Tris-HCl pH 7.5, 20 mM KCl) using a 30 kDa molecular weight cutoff Centricon centrifugal filter (Amicon), which improved transcription efficiency. The template sequence was verified by Sanger sequencing. The DNA template was effectively labeled by duplex formation with a 20 % molar excess of labeled DNA oligonucleotide, heating to 68 °C for 5 min and slow cooling to room temperature. The labeled template DNA duplex was not further purified, and the labeling efficiency (~100%) was quantified on a 4% agarose gel.

**Ensemble in vitro transcription experiments**. A stalled transcription elongation complex, composed of a DNA template, an RNAP molecule and a nascent RNA of 50 nucleotides (Fig. 1a), was formed by incubation of 25 nM (labeled) DNA transcription template, 100 μM ACU trinucleotide (Dharmacon), 10 μM ATP, CTP and UTP, 2 mM DTT, and 100 nM E. coli RNAP (NEB) in a buffer containing 50 mM Tris (pH 8.0), 14 mM MgCl$_2$, 20 mM NaCl, 0.04 mM EDTA, 40 μg ml$^{-1}$ BSA, 0.01% Triton X-100, for 20 min at 37 °C. Then, to prevent transcription reinitiation and to disrupt non-specific RNAP/DNA interactions that could lead to the loading of more than one RNAP molecule per DNA template, 1 mg ml$^{-1}$ heparin (or 20 μg ml$^{-1}$ rifampicin) was added, followed by incubation for a further 20 min at 37 °C. We verified the presence of a single RNAP molecule per DNA template in the stalled transcription-elongation complex (Supplementary. Figure 1b) using native gel electrophoresis. This is in agreement with the closest contacts between two RNAP molecules on the same DNA template being around 63 bp[61]. To test the hybridization efficiency of a biotinylated DNA oligo (Supplementary Fig. 1c,d) to the 5'-end of the nascent RNA for immobilization of the stalled transcription complex, 20 nM biotinylated DNA oligo was added simultaneously with the heparin. The biotinylated DNA oligo was obtained by autosticky PCR similarly to the transcription template.

Transcription elongation was initiated by simultaneous addition of 1 mM NTPs, 40 nM labeled DNA oligonucleotide complementary to the 5'-end or 3'-end of the nascent RNA and 150 mM KCl. After 20 min, sucrose was added to a 6% (w/v) final concentration and the samples were immediately loaded on a 2–4% agarose gel. The fluorescent bands were visualized and quantified using a VersaDoc$^{TM}$ MP 4000 Imager (Bio-Rad).

For denaturing PAGE experiments, transcriptions were performed identically except that during stalled complex formation, 150 nM [α-$^{32}$P]-ATP was added and the labeled DNA oligo binding to the 5'/3'-end of the nascent RNA was omitted. Furthermore, the reactions were stopped by addition of urea to 4 M before loading on an 8 % denaturing PAGE gel. The gels were visualized on a Storm 820 Molecular Imager (GE Healthcare).

**S15 protein cloning, expression and purification**. S15 Escherichia Coli full-length ribosomal gene (NCBI: NC_000913.3) with introduced surface accessible cysteine residue (T79C mutation) was cloned into pRSF-1b vector (Novagen) without any purification tags by Gibson assembly cloning. For S15 protein production, the pRSF1b-S15 plasmid was expressed in the E. coli BL21-Gold (DE3) strain. Cells were cultivated with vigorous shaking at 37 °C in LB medium with Kanamycin (50 μg ml$^{-1}$), and protein expression was induced with 1 mM IPTG when the culture reached an optical density OD$_{600}$ of 1.0 and incubated for another 5 h at 37 °C with shaking. Cells were harvested by centrifugation at 4000 rpm and then resuspended in sonication buffer (20 mM Tris-HCl pH 7.0, 0.1 M NaCl). The mixtures were lysed with ultrasound (10 s ON, 50 s OFF, for 5 min) at 0 °C and then spun down at 14,000 r.p.m. for 20 min. Inclusion bodies were washed with 0.5% (v/v) Triton X-100 and then spun down at 12,000 r.p.m. for 20 min. The Triton-X-100 washing was repeated 3 times. The pellets were solubilized in 50 ml of 8 M guanidine chloride in 50 mM Tris-HCl, pH 7.0 and incubated overnight at 4 °C with gentle agitation. The protein mixture was centrifuged and the supernatant dialyzed against 6 M urea, 50 mM Tris-HCl pH 7.0. The fraction of precipitated protein was removed by centrifugation. Purification of the S15 protein was performed under denaturing conditions in 6 M urea. The protein mixture was passed over a 5-ml SP HiTrap column (GE Healthcare), equilibrated with 6 M urea in 20 mM Tris-HCl pH 7.0 (buffer A) and eluted with a NaCl gradient of buffer B (buffer A, containing 1 M NaCl).

The S15 protein was refolded by dialysis against 20 mM Tris-HCl pH 7.6, 20 mM NaCl, 0.5 mM EDTA, and 0.5 mM DTT (buffer C) overnight at 4 °C. The soluble mixture of the refolded protein was passed over a 5-ml heparin HP column in buffer C and eluted with a NaCl gradient of buffer D (buffer C, containing 1 M NaCl).

The S15 protein was further purified by size-exclusion chromatography on a Superdex 75 26/600 HiLoad gel filtration column (GE Healthcare) equilibrated with 0.1 M NaCl, 0.5 mM EDTA, 0.5 mM DTT, and 20 mM Tris-HCl pH 7.6. The monomeric fractions were collected and concentrated using a Vivaspin 2 kDa MWCO centrifugal filter (Sartorius). The protein concentration was quantified with amino acid hydrolysis assay (AAA) at the UC Davis Molecular Structure Facility Core.

**S15 protein fluorescence labeling**. To reduce the protein cysteine residue, 1 mg of S15 was incubated for 2 h with 10 mM DTT on ice. Then, ammonium sulfate powder was added to the protein solution to a final concentration of 70% (w/v) and incubated for 20 min at 4 °C with gentle agitation. The precipitated protein was separated by centrifugation at 14,000 r.p.m. for 20 min, washed with 70% (w/v) ammonium sulfate in labeling buffer A (100 mM Na$_2$HPO$_4$/KH$_2$PO$_4$ phosphate buffer pH 7.0, 100 mM NaCl) and spun down again at 14,000 r.p.m. The washed pellet was dissolved in 475 μl labeling buffer, containing 6 M urea (labeling buffer A1).

A concentration of 1 mg of Cy5 maleimide (GE Healthcare) was dissolved in 25 μl ultra-pure DMSO (Sigma) and mixed with the protein solution. Dye conjugation was performed for 30 min with moderate shaking at RT and the reaction was quenched by addition of a final concentration of 0.5 % (v/v) β-mercaptoethanol.

The non-conjugated dye was removed from the labeled S15 protein using 5 ml Nap5 columns (GE Healthcare) equilibrated with buffer A1, loading a third of the reaction mixture per column (170 μl). The fractions containing labeled S15 protein were determined by a 4–20% SDS-PAGE gel visualizing the Cy5 fluorescence on a VersaDoc™ MP 4000 Imager (Bio-Rad) and the protein by Coomassie staining. To further remove the non-conjugated dye, the labeled S15 protein fractions were pooled, refolded by 10-fold dilution with 20 mM Tris-HCl pH 7.6, 20 mM NaCl, 0.5 mM EDTA (buffer E) and passed over a 1 mL heparin HP column (GE Healthcare) at 4 °C. The column was washed with 10 column volumes of buffer E and the labeled protein was eluted with 1 M NaCl in 20 mM Tris-HCl pH 7.6, 0.5 mM EDTA (buffer F). The protein was stored at -80 °C until usage.

To determine the protein Cy5 labeling efficiency (~95%), the dye-coupled protein concentration was calculated based on the extinction coefficient of 250,000 M$^{-1}$ cm$^{-1}$ for the Cy5 dye at 650 nm and subtracting the amount of free Cy5 dye (<5%) quantified on a polyacrylamide gel. Labeling efficiency was determined using the ratio between dye-coupled and total protein concentration obtained from AAA.

**ZMW instrumentation and single-molecule imaging**. Single-molecule experiments were conducted with a commercial PacBio RS II sequencer that has been

modified to allow the collection of single-molecule fluorescence intensities from individual ZMW wells in four different dye channels corresponding to Cy3, Cy3.5, Cy5, and Cy5.5[20]. The RS sequencer uses two lasers for dye excitation at 532 nm and 642 nm. In all experiments, data were collected at ten frames per second for 10 min. The energy flux of the green laser was 0.30 μW μm$^{-2}$, and the red laser was at 0.24 μW μm$^{-2}$ when both lasers were used, or 0.32 μW μm$^{-2}$ if only the green laser was used for FRET experiments.

For single-molecule experiments, we immobilized the stalled transcription complexes at the bottom of biotin-functionalized ZMWs coated with NeutrAvidin, in one of two ways. One strategy entails immobilization by hybridizing a biotinylated DNA oligonucleotide to the 5'-end of the nascent RNA of the stalled complex (Fig. 1a). The biotin-DNA/nascent RNA hybridization efficiency is close to 100 % (Supplementary Fig. 1c, d) and immobilization is specific for the biotinylated stalled complex (Supplementary Fig. 1e). Alternatively, stalled transcription elongation complexes can be immobilized via a 5'-biotin moiety on the nascent RNA (Supplementary Fig. 1e), by priming transcription with a biotin-ACU trinucleotide (Supplementary Fig. 1a), followed by removal of the excess biotin-ACU using a size exclusion column. While hybridization of a biotin-DNA oligonucleotide to the nascent RNA is experimentally simpler, the latter approach would allow immobilization also in the presence of RNase H or helicases, for example when using whole cell extract.

A stalled transcription complex with a biotinylated DNA tether or biotin-ACU moiety was prepared as described for the ensemble experiments. 3–11 nM stalled complex was then incubated with 200 nM NeutrAvidin in 50 mM Tris (pH 7.5), 14 mM MgCl$_2$, 20 mM NaCl, 0.04 mM EDTA, 40 μg ml$^{-1}$ BSA, 0.01% Triton X-100, 2 mM spermidine, 1 mM putrescine and 150–500 mM KCl for 10 min at room temperature. The stalled complex concentration was adjusted to obtain an optimal SMRT Cell loading efficiency of ~ 15 %. Larger constructs required higher concentrations of stalled complex for optimal immobilization. In parallel, a SMRT Cell V3 ZMW chip (Pacific Biosciences) was prepared by first wetting the ship with Tris buffer (20 mM Tris (pH 8.0), 50 mM NaCl), then washing with the buffer used to incubate the stalled complex with NeutrAvidin, and finally incubating the stalled complex/NeutrAvidin mixture on the chip. This incubation required 20 min to several hours, in proportion to template length, to obtain a satisfactory immobilization efficiency. After immobilization, non-immobilized complex was removed by washing with reaction buffer, composed of 50 mM Tris (pH 7.5), 14 mM MgCl$_2$, 20 mM NaCl, 0.04 mM EDTA, 40 μg ml$^{-1}$ BSA, 0.01% Triton X-100, 2 mM spermidine, 1 mM putrescine, 150 mM KCl, 0.25% Biolipidure 203, 0.25% Biolipidure 206, 0.5 mg ml$^{-1}$ yeast total tRNA, 0.5% (v/v) Tween-20, and 0.1 mg ml$^{-1}$ casein. The wash buffer also contained an oxygen-scavenging system consisting of 2.5 mM protocatechuic acid (PCA) with 250 nM protocatechuate-3,4-dioxygenase (PCD)[62], and 2 mM TSY (Pacific Biosciences) to minimize fluorescence instability. After washing, 20 μl of this wash mix was left on the chip to keep the surface wet.

The delivery mix, which was robotically introduced during the single-molecule experiment, consisted of reaction buffer supplemented with rNTPs, 100 nM labeled DNA oligonucleotide for hybridization to the 3'-end of nascent RNA, and (where applicable) Cy5-labeled S15 protein. DNA oligonucleotides (IDT) were labeled at their 3'-end and for some experiments, a BHQ-2 quencher was attached to the other terminus[17]. We used Cy3-oligos with and without quencher in our experiments, but found a slightly reduced background fluorescence in presence of the quencher. Before the experiment, the SMRT cell was loaded into a customized PacBio RS II sequencer (Pacific Biosciences)[20]. At the start of the elongation experiment, the instrument illuminates the SMRT cell with a green laser (532 nm) and then automatically delivers 20 μl of the delivery mixture onto the cell surface at t ~10 s. All experiments were performed at 20 °C, except otherwise stated.

For detecting true co-transcriptional protein binding, a Cy3-labeled oligonucleotide was hybridized to the 5'-end of the nascent RNA (Fig. 6a). Because the 5'-end of the central domain RNA is located at the active site of the RNA polymerase in the stalled transcription elongation complex, the hybridization region for the Cy3-labeled oligonucleotide lies within the RNA exit tunnel of the RNA polymerase and is therefore not accessible for hybridization. Therefore, after immobilization of the stalled transcription complex to the bottom of the ZMWs and removal of the excess NTPs, we added another limited set of NTPs, omitting UTP, in order to walk the polymerase until it stalls at the next uridine nucleotide. Then, the Cy3-labeled oligonucleotide could be hybridized to the 5'-end of the nascent RNA in the stalled transcription complex. The real-time single-molecule experiment was initiated by the addition of NTPs and Cy5-S15. All the other conditions were identical to the standard experiment described above.

For the single-molecule experiments with protein binding to pre-transcribed and pre-folded RNA, we performed a similar procedure as for the co-transcription experiments except that rather than forming a stalled transcription complex by adding only 3 out of the 4 NTPs, we performed a transcription reaction using all 4 NTPs for 10 min at 37 °C, to generate the full-length RNA. Then, we incubated the RNA for 10 min at 37 °C with 1 mg ml$^{-1}$ heparin and 20 nM biotinylated DNA tether. The full-length RNA was diluted 4-fold and further incubated for 10 min at 37 °C with 200 nM NeutrAvidin in 50 mM Tris (pH 7.5), 14 mM MgCl$_2$, 20 mM NaCl, 0.04 mM EDTA, 40 μg ml$^{-1}$ BSA, 0.01% Triton X-100, 2 mM spermidine, 1 mM putrescine and 150 mM KCl and 100 nM Cy3-oligo binding to the 3'-end of the full-length RNA. Except for a shorter 10 min immobilization time, immobilization was identical to the co-transcriptional experiments as was delivery

mix injection during the ZMW experiment except that the Cy3-oligo and the NTPs were omitted in the delivery mix.

**Data analysis.** Single-molecule data were analyzed with in-house–written MATLAB (MathWorks) scripts[20,63]. Only traces showing a fluorescence intensity increase or decrease, which is characteristic for transcription elongation, and with the subsequent binding of a single Cy3-labeled DNA oligonucleotide, which is binding to the 3'-end of the nascent RNA and indicates the presence of a single full-length RNA, were selected for further analysis.

To determine the exact start and end of transcription elongation, we first manually picked regions of constant fluorescence intensity preceding ($I(z1)$ in Fig. 2e) and following transcription elongation, $I(z2)$. We then fitted a horizontal line to those regions with the intercept of those fits corresponding to $I(z1)$ or $I(z2)$, respectively. Next, we fitted the fluorescence intensity increase during transcription elongation to a single exponential function, $y = a \times \exp(-b \times t) + c$ (Fig. 2e and see also Supplementary Note 2 and Supplementary Fig. 2 for discussion about the exponential fitting model). The intersection points of the two horizonal lines with the exponential fit correspond to the start and end of transcription, respectively (Fig. 2e). Representative fits are shown in Supplementary Fig. 8. Depending on the signal to noise ratio of the trace and the length of the RNA construct, the determination of the exact moment for start and end of transcription can have an uncertainty of up to a 2–3 s. This translates to an uncertainty for the determination of the transcription rate of single RNA molecules and is more pronounced for smaller RNA molecules (<200 nt). As an example, a specific RNA molecule of 200 nt with a transcription time of 10 s (Fig. 3b) has an average transcription rate of 20 nt s$^{-1}$. If the determination of the transcription time of this specific RNA molecule has an uncertainty of ± 2.5 s due to a low signal to noise ratio of that trace, this results in a range for the average transcription rate of 16-26.7 nt s$^{-1}$ for this specific RNA molecule. We note that with the large amount of data generated by the ZMW technology it is possible to select only traces with a high signal to noise ratio, thereby reducing the uncertainty in the determination of the transcription rate at the single-molecule level.

For the Cy5-S15 binding experiments, binding was detected by the presence of a high-FRET value between Cy5-S15 and the Cy3-oligo binding to the 5'-end or 3'-end of the nascent RNA (Fig. 1 and Fig. 5b). We calculated the FRET efficiency by $E_{FRET} = I_A / (I_A + I_D)$ from background and spectral bleed-through corrected traces (Supplementary Fig. 9), where $I_D$ and $I_A$ are the apparent fluorescence intensities of the donor and acceptor, respectively. To assign the bound state, we defined the threshold in the middle of the two FRET states (Supplementary Fig. 9 and 10), with subsequent manual inspection for traces that showed substantial non-specific Cy5-protein binding and therefore a non-zero FRET efficiency baseline (Supplementary Fig. 11). Traces with substantial non-specific Cy5-S15 protein binding show a wavy Cy5 fluorescence baseline, resulting in a wavy FRET baseline (see Supplementary Fig. 11). For those traces, we manually had to adapt the FRET threshold to assign a bound state. We use the presence of a clear anti-correlated change in Cy3 and Cy5 intensities, which can be readily distinguished from non-specific Cy5-protein binding (see Supplementary Fig. 11a) due to the presence of a high-FRET value in the bound state by design (see Fig. 5b). In contrast, non-specific Cy5-protein binding results only in a Cy5 fluorescence increase in absence of a reciprocal decrease in Cy3 fluorescence. Alternatively, to assign anti-correlated Cy3-Cy5 intensities automatically, we use, instead of FRET, another metric defined by $I_A(t) * [I_D(t)-I_D(max)]^2$, where $I_D(max)$ is the intensity of the Cy3-oligo in absence of FRET. This metric allows us to assign automatically a bound state using a single threshold per trace even for traces with very strong non-specific Cy5-protein binding (see Supplementary. Fig. 11d–f).

All dwell times were fitted to single- or double-exponential distributions using maximum-likelihood parameter estimation in MATLAB. Cy5-S15 binding was defined as kinetically stable binding if the protein-bound lifetime was > 10 s. This threshold was based on the clustering of the ensemble of all traces into N-class and P-class traces visualized in a scatterplot of the longest S15-bound lifetime per trace versus the number of binding events per trace (Supplementary Fig. 6d).

**Simulations of relative fluorescence in ZMW aperture.** In order to model the relative fluorescence detected by the system as a function of the position in the metal aperture, we needed to determine the electric field strength generated by the excitation source as well as the emission efficiency of the fluorescence. These electromagnetic fields were modeled using finite-difference time-domain (FDTD) simulation tools (https://kb.lumerical.com/en/sp_fluorescence_enhancement.html). The emission rate of the dye is proportional to the electric field strength $|E|^2$ of the excitation. The laser source in the PacBio instrument is modeled as a single wavelength Gaussian beam with wavelength 532 nm and waist radius 1 μm. The fluorescence emission is modeled as the incoherent sum of three orthogonal dipole sources with wavelength range 590-630 nm. The optical collection path is modeled as a simple microscope with numerical aperture of 0.9[23]. The fluorescence detected is then the product of the excitation field strength and the relative efficiency of the incoherent sum of dipoles. We calculated this value for radial and azimuthal positions within a metal aperture with a radius of 70 nm and a thickness of 90 nm. We fitted the simulation results to the following exponential function: $I = I_0 \times \exp(-cz)$, where $c$ is a function of $r$ and $z$.

**Statistics and reproducibility**. Measurements from single-molecule fluorescence assays resulted from a specified number (n) of molecules from a single experiment.

## Data availability

Source data for Figs. 2f–h, 3a–d, 4c, 5e and Supplementary Figs. 1g–i, 2g–j, 3c, d, 6b,e,f are available as a source data file. A reporting summary for this Article is available as a Supplementary Information file. All the other experimental data are available upon reasonable request.

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

## Acknowledgements

The authors thank Edit Sperling, Rajan Lamichhane, Alexey Petrov, Rosslyn Grosely, John Hammond, Vadim Patsalo, Josh Silverman and the Williamson, Puglisi and David Millar laboratories for helpful discussions and experimental advice. We thank Alex Johnson and Junhong Choi for critical reading of the manuscript. This work was supported by the Swiss National Science Foundation (SNF) early postdoc.mobility grant no. P2EZP3-152131, advanced postdoc.mobility grant no. P300PA-160978 and Human Frontier Science Program grant no. LT000628/2015-L to O.D, NIH R01 GM053757 to J.R.W and NIH R01 GM051266 to J.D.P.

## Author contributions

O.D. designed the project, with input from J.R.W. and J.D.P. O.D. performed all the biochemical and single-molecule experiments and data analysis. G.A.S. cloned, expressed, purified and labeled proteins. A.G. performed the optical modeling in the ZMW. O.D. wrote the manuscript, with input from S.E.O'L, J.R.W. and J.D.P.

## Additional information

**Competing interests:** A.G. is an employee of Pacific Biosciences Inc., a company commercializing DNA sequencing technologies at the time that this work was completed. The remaining authors have no competing interests.

