## [Peer Review File · Nature Communications]

Reviewers' comments:

Reviewer #1 (Remarks to the Author):

Duss et al describe an innovative, state-of-the-art approach to studies of ribonucleoprotein complexes. The authors use zero-mode waveguide technology and single-molecule fluorescence measurements to concurrently follow transcription and protein binding. The authors demonstrate that the population of a newly transcribed 16S rRNA fragment is structurally heterogeneous and folds into at least three different conformations characterized by distinct kinetics of ribosomal protein S15 binding. The manuscript summarizes the impressive body of work and includes a sophisticated analysis of complex single-molecule kinetic data. This article will likely grab the attention of many scientists interested in nucleic acid-protein interactions and single-molecule methods. However, the authors should address a few issues before publication:

- 1) The manuscript is rich with information. The authors apparently tried to mitigate this by placing many essential aspects of data analysis and interpretation into Supplementary materials, which contains 9 "supplementary discussion" pieces. However, switching between the main text and these pieces makes it hard to follow the paper. Shortened versions of these discussion pieces should be included in the main text.
- 2) Since the manuscript describes an innovative experimental approach that can be applied to other systems, more details regarding data analysis need to be included. In Methods, the authors say that states in single-molecule traces were assigned manually (e.g. Fig. 1 d-e). What were the criteria chosen for state assignment? How did the authors distinguish real transitions from noise? This is particularly important since the authors analyze the kinetics of S15 binding based on the state assignments. How did the authors distinguish between spurious fluorescent signals typical for single-molecule fluorescent measurements and actual single-molecule traces without introducing bias in data selection? Furthermore, figure legends for some figures that examine populations of molecules do not show the number of traces used in the analysis (e.g. Fig. 2 f-h, Suppl. Fig 2, 3, 6). The authors should correct this. Although the authors state in the introduction that ZMW technology allows for the detection of "thousands of single biomolecules," Figures 3, 4 and 5 mostly show the analysis of less than 100 traces. Why were so few traces included in the analysis?
- 3) The authors used a fluorescently-labeled DNA oligo to detect S15 binding via FRET. How did the authors make sure that annealing of this oligo to a nascent transcript does not perturb S15 binding and RNA folding?

Reviewer #2 (Remarks to the Author):

Duss and co-workers describe a method for tracking the binding of fluorescently labeled proteins to single nascent RNA transcripts, using a zero-mode waveguide TIRF instrument. It is well known that RNA transcripts begin to fold during transcription, but this is an extremely complicated problem to study and to model. This manuscript reports a very promising and exciting approach to this problem. Duss et al monitor transcription by measuring the increase in fluorescence intensity as a labeled DNA template is pulled into the TIR illumination field by an immobilized transcription elongation complex. TIRF is also used to examine binding of the small subunit ribosomal protein S15 to co-transcriptionally folded 171 nt fragment of the E. coli 16S rRNA. This model system was extensively studied in the Williamson lab and is a good choice for the proof-of-concept experiments reported here.

The main findings are that S15 can bind the rRNA fragment shortly after it is transcribed. Many complexes are stable ("N class"), but particularly at low temperatures, a sizeable fraction of transcripts bind S15 protein only transiently ("P class"). Overall, a number of cutting-edge experiments indicate that this method can work. The experimental setup is well-conceived, and a large number of controls demonstrate that immobilized complexes are transcriptionally active and produce full length RNA that can be detected afterwards using a Cy3-labeled oligo. The authors have clearly done a tremendous amount of work to get the system to its current stage of development.

Nevertheless, I have substantial concerns regarding the specific data analysis and interpretation that I believe would require substantial revision to address. As outlined below, these concerns have to do with the accuracy of determining the start and end of transcription, which influences their calculation of transcription speed, and dye emission overlap and a lack of fluorescence background correction, which make it difficult or impossible to interpret FRET data. This uncertainty undermines the assignments of S15 binding and dissociation events.

Major concerns:

1. The Cy3.5 intensity gradually slopes up at the beginning of the experiments, before reaching a plateau (sometimes) and then falling sharply when the DNA dissociates. The use of two Cy3.5 dyes is a clever way to distinguish dissociation from photobleaching. It is not clearly stated in the manuscript, however, how they measure the exact moment elongation restarts after adding NTPs, which is hard to do from a sloping baseline. In addition, the traces are noisy near the end of the transcription, and the beginning of the fluorescence plateau indicating paused termination is often hard to distinguish in the traces that are shown (i.e. Fig. S7C, and Fig. 1d,e). Do the authors fit the traces with some type of hidden Markov model? or by eye? This should be better explained.
2. A related concern is whether the authors have the right model for the change in Cy3.5 intensity with transcription in the TIR evanescent field. The authors have carefully considered the relevant parameters, which they explain thoroughly in the supplement and Fig. S2, very much to their credit. In Fig. S2h, however, one can easily see that the measured ratio $y = \Delta I(\text{TK}) / \Delta I(\text{diss}) = -0.5$ when the fitted curve is extrapolated to $\langle R(l) \rangle = 0$, which is an unphysical result. This discrepancy suggests there is a zone of near constant fluorescence intensity near the end of transcription, when the DNA contour length passes below a certain threshold. (Another zone of constant fluorescence should occur at the beginning of transcription of long templates.) I agree that Fig. 2 and Fig. S2 show that the change in fluorescence intensity measures movement of the DNA through the field of illumination, but the model may not properly account for end effects that are important for determining the times of transcription restart and termination.
3. The authors report a transcription speed that is twice as fast as what has been observed by single molecule transcription experiments from the Gelles lab (12 nts/s; Yin et al. Biophys J. 1994), Block lab (~12 nts/s; Frieda et al. Science 2012), and Bustamante lab at 20 and 35 °C (~5 and 15 nts/s, respectively). Could the authors please comment on this difference and address how much this might have to do with how they measure the start and end of transcription?
4. The intensity of the DNA Cy3.5 signal is sometimes mirrored by the Cy5 channel, suggesting that there is significant bleed through (Figure 1d,e). Do the authors account for this, and if so, how? If not, it is unclear to me how they can determine protein binding in the Cy5 channel during transcription. The traces seem quite noisy due to low fluorescence signal overall.
5. None of the fluorescence intensity traces have been corrected for the background. This is essential for interpreting smFRET data and is very important for understanding how colocalization data were

assigned. I recommend that the authors reanalyze their data, correcting for the background, or illustrate the background signal as a separate trace if background correction is not feasible.

As an example, in Fig. 1d (N class), the Cy5 intensity first rises and falls in parallel with the Cy3.5 intensity, presumably due to bleed through. This is followed by several “wobbles” in intensity between 30 and 50 s, a transition to a somewhat bright state at 60 s, and finally a transition to a brighter state at 85 s, which the authors assign to S15 binding. The Cy3 intensity also drops at 85 s, as expected if there is FRET between Cy3 on the oligo and Cy5 on S15. How do the authors know that the earlier wobbles, which are more intense and have lower frequency than the noise, do not reflect a low FRET S15 bound state? At 300 s, the Cy5 intensity drops and the Cy3 intensity rises, presumably due to photobleaching or dissociation of S15. However, the Cy3 intensity doesn't rise to the level it exhibited at 60 s, before S15 is said to have bound, which it should do if S15 has dissociated or gone dark. This is puzzling.

7. The authors say they observe transient binding of S15 in the P class subset. In Figure 1e, how do the authors distinguish dissociation of S15 from fluctuations into a low FRET conformation? (Both scenarios would produce anti-correlated changes in Cy3 and Cy5 intensity; it can be hard to distinguish zero FRET from low FRET even with background correction. Maybe one could test whether P class S15 molecules exchange with free S15.) Additionally, the trace illustrating the P-class in Supp. Figure S5 is barely above the noise. The anti-correlated peaks indicate that FRET is occurring. Since the background is not shown, however, it is difficult to determine whether this is legitimate binding. The figure legend indicates “high FRET”, yet this cannot be said unless the traces are appropriately background corrected.

Minor points:

8. The labeling schemes outlined in Fig 2a-d are wonderful and clearly a lot of work has been done to characterize this system. However, the figure should be adjusted to be more easily understood (i.e. bigger traces) and there needs to be more discussion on each labeling scheme in the legend and in the text. For example, experiments in which both the DNA and the oligo are labeled with the same dye are initially confusing unless one carefully examines the cartoons for each panel.

9. Please indicate the time when NTPs were injected in each trajectory; if this equals time ‘zero’, please state this somewhere. Also, is there a substantial delay between the addition of NTPs and the restart of transcription? I would expect transcription restarts in less than 1 s, but perhaps this is not the case.

10. The 50 nt initial transcribed sequence used to generate the stalled elongation complexes and the long promoter region would allow for at least 2 RNAP to load onto the DNA template (based on the 23-25 bp footprint of E. coli RNAP). Could the authors note somewhere that this is a possibility and that it would complicate detection of protein binding during transcription?

11. Suppl. Discussion 6. “...Cy3 dyes are predicted to clash with the RNA entry channel...” should be DNA entry channel.

12. I found the presentation of the S15 binding kinetics a bit confusing. The on-rate is slower than diffusion controlled; is this because binding is multi-step? Could multi-step binding explain why the observed on-rates do not increase linearly with S15 concentration? Or, do the authors think that the slower observed binding rate reflects a misfolded RNA population? It should be possible to account for the latter by separately analyzing N class and P class binding kinetics. Also, does Fig. S6f report the fast or slow observed binding rate?

13. In Supp Figure S7, there is a "FRET" signal in Cy5 before the Cy3 oligo has bound to the RNA and before the DNA template has dissociated. Is this bleed through from the Cy3.5 channel? The signal appears to increase without a mirrored increase in Cy3.5, so why would there be an increase in the Cy5 signal unless this was an S15 binding event? The authors assign S15 binding concomitant with Cy3 signal but there is not a significant increase in the fluorescence signal in Cy5, in fact it appears to slowly increase over the entire movie. There does not seem to be any anti-correlation until possibly 160 s, near the end of the trace. Is this typical?

14. Also in Figure S7c, what is the meaning of the three panels on the right? They don't seem to be connected with any other panel.

Reviewer #1:

1) The manuscript is rich with information. The authors apparently tried to mitigate this by placing many essential aspects of data analysis and interpretation into Supplementary materials, which contains 9 “supplementary discussion” pieces. However, switching between the main text and these pieces makes it hard to follow the paper. Shortened versions of these discussion pieces should be included in the main text.

We have included 610 additional words of shortened versions of supplementary discussion back into the main text. The additions are highlighted in green in the main text and are shown below:

Discussion 2 and Supplementary Fig. 2). Using a semi-empirical model, which treats the DNA as a worm-like chain (WLC)²⁴, we obtain an evanescent decay constant c , which is in agreement with calculations of the decay constant within the ZMW holes using a finite-difference time-domain approximation to Maxwell's electromagnetic equations (Supplementary Figs. 2c,d and Supplementary Discussion 1 and 2). Overall, the normalized

To mark the transcription end-point independently, a Cy3-labeled DNA oligonucleotide targeting the 3'-end of the full-length RNA was simultaneously added with the NTPs at the beginning of the experiment (Fig. 1b,c). In > 75 % of the single-molecule traces with transcription-dependent Cy3.5 intensity changes, a Cy3 signal appears shortly after template dissociation. The rate of Cy3 oligo binding to the nascent RNA can be determined by fitting an exponential function to the distribution of times between end of transcription marked by the end of the Cy3.5 fluorescence increase, and the binding of the Cy3 oligo probe. We found a fast bimolecular association rate ($5.2 \pm 0.1 \times 10^5 \text{ M}^{-1} \text{ s}^{-1}$

To modulate the RNAP stalling duration at the 3'-end of the DNA transcription template, these experiments were repeated with a construct containing a single *T7Te* transcription terminator directly followed by a single-strand break in the non-template strand and the two bulky Cy3 dyes attached on the opposite template strand (Supplementary Fig. 3b). In this latter construct, the RNAP is not expected to terminate efficiently because the Cy3 dyes are predicted to clash with the DNA entry channel, preventing the active site of the polymerase from reaching the transcription-terminating nucleotide (Supplementary Fig. 3b). While the fraction of molecules having very long RNAP stalling of several minutes (Fig. 4b) was similar to the triple terminator construct (5-15 %), the average residence of the fast dissociating RNA polymerase molecules at the blocked terminator was increased to 20 ± 5 seconds (Fig. 4c). These results demonstrate that the dissociation kinetics of the RNA polymerase are influenced by the structural context at the 3'-end of the DNA template. The fast dissociation kinetics observed at the triple terminator are in agreement

Those RNA molecules with bursts of transient S15 binding have at least a partially formed 3-way junction because S15 binding is entirely abolished in a G752C 3-way junction mutant; we do not detect transient S15 binding in that mutant. To obtain more direct structural information on the misfolding of those RNA molecules, we labeled H20 and H22 in the 3'-way junction construct with a Cy3 and Cy5 dye, respectively (Supplementary Fig. 5a,b). Ensemble and single-molecule experiments have demonstrated that S15 binding leads to closure of the H20/H22 helix conformation, which was manifested by a high-FRET state due to close proximity of the Cy3 and Cy5 dyes 38-40. Using this labeling scheme, we observed two types of FRET dynamics. The first type of trace, which represents the majority (> 80%) of the RNA molecules, showed a transition from a low-FRET state to a high-FRET state shortly after S15 protein delivery and that persisted for several minutes (Supplementary Fig. 5c). This behavior corresponds to the kinetic signature we found for the N-class molecules that showed kinetically stable S15 binding shortly after protein delivery. In addition, we found ~ 10 % of traces showing multiple transitions into the high-FRET state that persisted for only 1-2 seconds representing the kinetic binding behavior of S15 in the P-class RNA molecules (Supplementary Fig. 5d). Those data demonstrate that the RNA transitions into a docked conformation, as present in the native ribosome, both during the stable S15 binding events in the N-class molecules (Supplementary Fig. 5c) as well as during the transient binding events in the P-class molecules (Supplementary Fig. 5d). This multistep binding

compared to single-nucleotide resolution in optical trap experiments 45. RNAs significantly shorter than 100 nucleotides, will generate only a small intensity change during transcription progression, which may become difficult to detect. Yet, complementary to our approach, detection of transcription progression of very small RNAs (< 30 nucleotides) in real-time is possible by following a FRET efficiency change between a 3'-end-labeled DNA transcription template and a labeled RNAP 13,18. We readily observed S15 binding to

2) Since the manuscript describes an innovative experimental approach that can be applied to other systems, more details regarding data analysis need to be included. In Methods, the authors say that states in single-molecule traces were assigned manually (e.g. Fig. 1 d-e). What were the criteria chosen for state assignment? How did the authors distinguish real transitions from noise? This is particularly important since the authors analyze the kinetics of S15 binding based on the state assignments. How did the authors distinguish between spurious fluorescent signals typical for single-molecule fluorescent measurements and actual single-molecule traces without introducing bias in data selection?

We can provide the additional details, as also requested by reviewer #2 (further detailed in points 4,5 of reviewer #2). We have added several additional supplementary figures (S8-S12) and methods sections to provide this information. We explain in detail how we assign states, and how traces with considerable adsorption of protein to the chip were treated. The background and bleedthrough did not affect the conclusions, as anticorrelated fluorescence changes were used to determine FRET. Nevertheless, we have now corrected the traces for background and bleed-through and show the FRET efficiency for clear visualization of the state assignment. As expected, the results are essentially unchanged.

Furthermore, figure legends for some figures that examine populations of molecules do not show the number of traces used in the analysis (e.g. Fig. 2 f-h, Suppl. Fig 2, 3, 6). The authors should correct this.

We have added the number of molecules analyzed in the figure legends.

Although the authors state in the introduction that ZMW technology allows for the detection of “thousands of single biomolecules,” Figures 3, 4 and 5 mostly show the analysis of less than 100 traces. Why were so few traces included in the analysis?

Only a portion of the entire SMRT cell was analyzed because of the large amount of data generated and because of the time-consuming data evaluation. We typically picked 80-170 traces, depending on the complexity of the data evaluation, to have sufficient data for good statistics. This number of traces is a typical number also used for TIRF experiments or other ZMW studies. To obtain this number of traces we typically used between 1% and 50% of the entire SMRT cell depending on the complexity of the experiment. In particular, the immobilization efficiency of the stalled transcription elongation complex is less efficient for longer constructs, limiting the number of useful traces that can be obtained. In addition, the immobilization efficiency varies over the location on the SMRT cell, and the SMRT cell batch used.

In order to demonstrate the large amount of data generated during a single experiment with the ZMW, we have analyzed the transcription of > 2000 RNA molecules on a single chip during a single experiment and have added two supplementary Fig. panels (Supplementary Fig. 1h,i), showing the distribution of transcription times and the location of the 2056 analyzed RNA molecules on the SMRT cell. In addition, we have added a reference to this supplementary figure in the main text (underlined):

“...We present here a highly-multiplexed approach for monitoring transcription of thousands of single RNA molecules in real time during one experiment (Supplementary Fig. 1h,i),....”

3) The authors used a fluorescently-labeled DNA oligo to detect S15 binding via FRET. How did the authors make sure that annealing of this oligo to a nascent transcript does not perturb S15 binding and RNA folding?

We agree this is a potential concern, but the following arguments and results support the conclusion that S15 binding and nascent RNA folding are not perturbed by the fluorescently-labeled DNA oligo (Cy3-oligo):

- The Cy3-oligo is complementary to non-ribosomal sequences that were added directly adjacent to either the 5'- or 3'-end of the native 3'-way junction RNA. By design, the Cy3-oligo has less than 5 consecutive base-pairs with any part of the native RNA in order to minimize even transient binding of the Cy3-oligo with the nascent RNA. The hybridization lifetimes of 7bp DNA/DNA or 6bp RNA/RNA duplexes are in the order of 0.2 seconds or 10 seconds, respectively (Cisse & Ha, NSMB, 2012) and 7bp RNA/DNA duplexes are estimated to have a bound-lifetime of around 5 seconds (Johnson-Buck & Walter, Nature Biotechnology, 2015). Thus, a potential base-pairing of < 5bp between the DNA Cy3-oligo and the nascent RNA would have minimal stability and therefore minimal potential to influence RNA folding.
- Hybridizing Cy3-oligos with different sequences to the artificial regions flanking either the 3'-end or the 5'-end of the nascent RNA show similar populations of the three folding classes and S15-bound lifetimes suggesting that RNA folding and S15 binding is not dependent on the sequence of the Cy3-oligo, further demonstrating no effect of the Cy3-oligo on S15 binding and RNA folding.
- Pre-folding the RNA and pre-hybridizing the Cy3-oligo to the flanking regions before immobilization of the RNA to the ZMW surface, buffer exchange and the subsequent real-time monitoring upon addition of the S15 protein also shows all three RNA folding populations with similar S15 binding kinetics as the co-transcriptionally folding RNAs demonstrating that RNA folding and protein binding are not affected by Cy3-oligo addition.
- Using the shorter construct in which the RNA is labeled with a Cy3 and Cy5 dye to monitor RNA conformational changes induced by binding of unlabeled S15 (in absence of Cy3-oligo) shows all three RNA folding classes (see suppl. Fig S5).

- Finally, in a separate study using a different RNA and protein (unpublished results), we found that addition of a combination of 5 different unlabeled DNA oligos with random sequences at a concentration of 2uM each (total 10uM compared to 100nM used for the Cy3-oligo in the present study), we did not find a change in RNA folding suggesting that the unspecific binding of DNA oligos do not influence nascent RNA folding efficiency.

Overall, these observations strongly suggest that RNA folding and S15 protein binding are unaffected by the presence of 100nM Cy3-oligo.

Reviewer #2:

1. The Cy3.5 intensity gradually slopes up at the beginning of the experiments, before reaching a plateau (sometimes) and then falling sharply when the DNA dissociates. The use of two Cy3.5 dyes is a clever way to distinguish dissociation from photobleaching. It is not clearly stated in the manuscript, however, how they measure the exact moment elongation restarts after adding NTPs, which is hard to do from a sloping baseline. In addition, the traces are noisy near the end of the transcription, and the beginning of the fluorescence plateau indicating paused termination is often hard to distinguish in the traces that are shown (i.e. Fig. S7C, and Fig. 1d,e). Do the authors fit the traces with some type of hidden Markov model? or by eye? This should be better explained.

We have added an additional methods section (see changes underlined), have modified Fig. 2e and have added a new supplementary figure S8 with representative traces to demonstrate how we rigorously determine the exact moments for the start and end of transcription. Note that the traces mentioned by the reviewer have been background and bleed-through corrected and are now presented in Supplementary Figures S11e (S7c before), S11d (before Fig. 1d) and S11a-c (before Fig. 1e).

“...only traces showing a fluorescence intensity increase or decrease, which is characteristic for transcription elongation, and with the subsequent binding of a single Cy3-labeled DNA oligonucleotide that is binding to the 3'-end of the nascent RNA and indicates the presence of a single full-length RNA, were selected for further analysis. To determine the exact start and end of transcription elongation, we first manually picked regions of constant fluorescence intensity preceding $I(z_1)$ in Fig. 2e) and following transcription elongation, $I(z_2)$. We then fitted a horizontal line to those regions with the intercept of those fits corresponding to $I(z_1)$ or $I(z_2)$, respectively. Next, we fitted the fluorescence intensity increase during transcription elongation to a single exponential function, $y=a*\exp(-b*t)+c$ (Fig. 2e and see also Supplementary Discussion S2 and Supplementary Fig. S2 for discussion about the exponential fitting model). The intersection points of the two horizontal lines with the exponential fit correspond to the

start and end of transcription, respectively (Fig. 2e). Representative fits are shown in Supplementary Fig. S8.

Furthermore, we have also referenced Supplementary Fig. S8 in the main text: “Consistent with a transcription elongation-dependent movement of the template 3'-end into the excitation volume close to the ZMW surface (Fig. 1c), we observed a monotonic increase in the Cy3.5 signal after NTP delivery (Fig. 1d,e and Supplementary Fig. S8).”

2. A related concern is whether the authors have the right model for the change in Cy3.5 intensity with transcription in the TIR evanescent field. The authors have carefully considered the relevant parameters, which they explain thoroughly in the supplement and Fig. S2, very much to their credit. In Fig. S2h, however, one can easily see that the measured ratio $y = \Delta I(\text{TK}) / \Delta I(\text{diss}) = -0.5$ when the fitted curve is extrapolated to $\langle R(l) \rangle = 0$, which is an unphysical result. This discrepancy suggests there is a zone of near constant fluorescence intensity near the end of transcription, when the DNA contour length passes below a certain threshold. (Another zone of constant fluorescence should occur at the beginning of transcription of long templates.) I agree that Fig. 2 and Fig. S2 show that the change in fluorescence intensity measures movement of the DNA through the field of illumination, but the model may not properly account for end effects that are important for determining the times of transcription restart and termination.

We are pleased that the reviewer appreciates our efforts to quantify significant fluorescence changes during transcription with a semi-empirical model. We felt that it was important to understand the physical basis for the observed effect. However, the details of the model and the underlying physical basis do not impact our ability to monitor the start/stop of transcription using the observed fluorescence changes.

As we note in the supplementary discussion, the complexity of the system required the introduction of an additional empirical parameter $\langle d \rangle$ in order to fit our data to the expected exponential form, and as we clearly state, we cannot extrapolate outside our experimental range of ~150-550 transcribed RNA nucleotides. We do not have a clear explanation for the physical meaning of the parameter $\langle d \rangle$. We have extended the supplementary discussion, and have added an additional supplementary figure panel presenting an alternative simpler empirical model, with the ratio $y = \Delta I(\text{TK}) / \Delta I(\text{diss}) = 0$ where the fitted curve is extrapolated to zero transcribed RNA length (Supplementary Fig. S2j).

Furthermore, we would like to point out that the data used to generate this model is independent of the determination of start and end of transcription, and depends only on the average (constant) intensity before and after transcription progression. Finally, we are not using this model to determine the start and end of transcription. These data demonstrate that the relative (normalized) intensity change during transcription within the ZMW wells is a defined quantity, which is independent of the transcribed RNA

molecule for a given length and is a defined (semi-empirical) function of the transcribed RNA length.

We strongly feel that a full understanding of this effect is well beyond the scope of this study, and note that the physical understanding of the phenomenon is not required to allow us to extract transcription start and stop times.

We have added following text to the supplementary discussion S2 (underlined):

Within our experimental range of ~ 150-550 transcribed RNA nucleotides, the experimental data can be well described by fitting to this modified equation (Supplementary Fig. 2i), and we obtain a decay constant c , which is in agreement with calculations of the decay constant within the ZMW holes using a finite difference time domain approximation to Maxwell's electromagnetic equations (Supplementary Figs. 2c,d and Supplementary Discussion 1). We note that this semi-empirical model does not appropriately describe the transcription of smaller RNAs extrapolated to zero length, partly because the assumption that RNA extension during transcription is independent of the transcribed RNA length, which is represented by the constant $\langle d \rangle$, does not hold anymore for small RNA molecules.

We find that a simpler empirical exponential model, which does not describe the DNA as a WLC but describes it as an extended rod, fits our data similarly well and also better describes the transcription of smaller RNAs extrapolated to zero length (Supplementary Fig. S2j). However, this simplified model yields a fluorescence decay constant, which is less in agreement with our calculations of the decay constant within the ZMW holes using a finite difference time domain approximation to Maxwell's electromagnetic equations.

Overall, a semi-empirical model, which treats the DNA as a WLC and includes an additional transcribed-RNA-length-independent parameter, describes our data well within our experimental range of ~ 150-550 transcribed RNA nucleotides and also correctly predicts the decay constant within the ZMW (Fig. 2h). Alternatively, the data can also be well-described by a simpler empirical model using a single exponential function with a single parameter (Suppl. Fig. S2j).

3. The authors report a transcription speed that is twice as fast as what has been observed by single molecule transcription experiments from the Gelles lab (12 nts/s; Yin et al. Biophys J. 1994), Block lab (~12 nts/s; Frieda et al. Science 2012), and Bustamante lab at 20 and 35 °C (~5 and 15 nts/s, respectively). Could the authors please comment on this difference and address how much this might have to do with how they measure the start and end of transcription?

Generally, we feel our transcription rate is well within the observed and expected rates observed using a variety of disparate methods. Our transcription rates are indeed slightly higher than in some previous single-molecule experiments using either optical tweezer experiments (Mejia & Bustamante, JMB, 2008; Frieda et al. Science 2012) or

tethered particle motion (Yin et al. Biophysical J. 1994). However, a recently published study using magnetic tweezers (Janissen & Meyer, Cell, 2018) established transcription rates of ~25 nts/sec at 20C and an alternative recently published single-molecule approach using modulation interferometry (Wang & Pertsinidis, Cell, 2016), actually demonstrates even higher transcription rates than we observe (~36 nt/s at 25 C and ~60 nt/s at 35 C). The approach by Wang & Pertsinidis and our approach require the tethering of only one component of the transcription elongation complex for immobilization (RNAP in Wang et al. and nascent RNA in our study). In contrast, in optical tweezer experiments, the transcription elongation complex is held between two optical traps and a link is made between the RNAP and the nascent RNA or the downstream end of the DNA. At the same time, a force is applied between the two anchor points, which could affect transcription speed. Similarly, in tethered particle motion, the RNAP is attached to a surface and a large micrometer sized bead is attached to one end of the DNA, which could also influence the transcription elongation dynamics.

Our transcription rates (and of Wang & Pertsinidis and Janissen & Meyer, Cell, 2018) are closer to the reported *in vivo* transcription rates. Also, transcription rates of rRNA (as we are using) are faster than for other RNAs (Chen & Xie, Mol. Syst. Biol., 2015). Those findings indicate that transcription rates vary depending on the experimental conditions (buffers, RNA constructs and more importantly experimental design such as how many anchor points are used to restrain the transcription elongation complex for immobilization).

To check if our transcription rates could be significantly overestimated because of a wrong estimation of start and end transcription, we assumed that transcription starts immediately upon NTP injection (assuming no mixing delay) and that transcription completes upon DNA template dissociation (assuming no stalling of the RNA polymerase at the 3'-end terminator). This corresponds to the longest possible transcription time (and the slowest possible transcription rate). We find that for our 517nts construct, this takes on average 30sec giving a lowest possible transcription rate of 17nts/sec. Thus, our transcription rates must be higher than reported in earlier optical trap or tethered particle motion experiments but are similar to a recently published magnetic trap study (Janissen & Meyer, Cell, 2018) and lower than reported using single-molecule modulation interferometry (Wang & Pertsinidis, Cell, 2016).

Overall, our transcription rates are in the range found with other single-molecule approaches.

We have already mentioned in the text:

“... with values comparable to previous bulk *in-vitro* transcription and single-molecule studies (10-36 nt s⁻¹), (Ref 11,25-28).”

And:

“...This is in good agreement with a previous single-molecule study using optical tweezers reporting 9.7 ± 0.7 kcal mol⁻¹ (Ref 29) and with the reported activation energy for the rate of RNA elongation *in vivo* of ~ 10.9 kcal mol⁻¹ (Ref 27).”

We have now also added the reference of Janissen & Meyer, Cell, 2018 that was published since our submission.

4. The intensity of the DNA Cy3.5 signal is sometimes mirrored by the Cy5 channel, suggesting that there is significant bleed through (Figure 1d,e). Do the authors account for this, and if so, how? If not, it is unclear to me how they can determine protein binding in the Cy5 channel during transcription. The traces seem quite noisy due to low fluorescence signal overall.

There is a defined bleed-through from the Cy3.5 to the Cy5 channel while bleed-through from Cy5 to Cy3.5 is minimal. Therefore S15-Cy5 protein stickiness from the Cy5 channel does not bleed-through into the Cy3.5 channel minimally affecting detection of transcription. On the other hand, the bleed-through from Cy3.5 into Cy5 has been described in Chen & Puglisi, PNAS, 2014 and is constant over the entire trace. Therefore, we can subtract the bleed-through as we now show in supplementary Fig. S9 and S10. We have also subtracted the bleed-through in all the relevant traces of the paper (see e.g. Fig. 1d,e mentioned by the reviewer). We also note that, because protein binding is only detected by a strong FRET between Cy3 and Cy5 (see point 5 reviewer #2 for more details), even with an imperfect Cy5 baseline during transcription (for traces with high S15cy5 stickiness) a specific binding event can be clearly distinguished from nonspecific association.

5. None of the fluorescence intensity traces have been corrected for the background. This is essential for interpreting smFRET data and is very important for understanding how colocalization data were assigned. I recommend that the authors reanalyze their data, correcting for the background, or illustrate the background signal as a separate trace if background correction is not feasible.

As an example, in Fig. 1d (N class), the Cy5 intensity first rises and falls in parallel with the Cy3.5 intensity, presumably due to bleed through. This is followed by several “wobbles” in intensity between 30 and 50 s, a transition to a somewhat bright state at 60 s, and finally a transition to a brighter state at 85 s, which the authors assign to S15 binding. The Cy3 intensity also drops at 85 s, as expected if there is FRET between Cy3 on the oligo and Cy5 on S15. How do the authors know that the earlier wobbles, which are more intense and have lower frequency than the noise, do not reflect a low FRET S15 bound state? At 300 s, the Cy5 intensity drops and the Cy3 intensity rises, presumably due to photobleaching or dissociation of S15. However, the Cy3 intensity doesn't rise to the level it exhibited at 60 s, before S15 is said to have bound, which it should do if S15 has dissociated or gone dark. This is puzzling.

This is a very important point raised by the reviewer and we added several supplementary figures, performed an additional experiment and wrote a new methods sections to address these issues. As we clearly mention in the manuscript at several places, we exclusively use FRET to DETECT protein binding using a STRONG FRET between the Cy3 oligo and the Cy5-labeled protein. However, we do not interpret smFRET data for RNA conformational changes because we have non-specific Cy5-S15 stickiness at higher protein concentrations, which complicates background correction for certain traces. This strong FRET signal allows us to clearly distinguish Cy5 protein binding (strong anticorrelated Cy3 and Cy5 intensity changes) from Cy5-protein stickiness (no concomitant anticorrelated changes in Cy3 intensity). The Cy3-oligo binding site has been designed such that there is a close (43 Å) distance between Cy5-S15 and the Cy3-oligo.

The inability of the 3-way junction mutant (G752C) to bind S15 and real-time monitoring of the RNA conformational changes upon S15 binding (Supplementary Fig. S5 and Supplementary Discussion 9, which is now moved to main text) show that S15 can only bind if the 3-way junction is formed and thus, S15 binding has to be observed as a strong FRET. We note that even if the 3-way junction is in the open form, this distance should not change significantly given our construct design (see Fig. 5b). Thus, S15 binding should always be detected as a strong FRET. Nevertheless, to rule out entirely that we are missing binding of S15 in a lowFRET or zeroFRET state, we have performed an additional experiment with direct Cy5-S15 detection (discussed further below).

As stated above, we don't believe that either the bleedthrough or background correction have affected our results. To demonstrate this we have now performed the analysis using background and bleedthrough correction (see also point #4 of reviewer #2). We now also show the calculated FRET efficiency. Finally, we have done four additional suppl. Figs. and added a new methods section, explaining in detail how we assign a Cy5-S15 bound state for normal traces and also for traces with very strong protein stickiness. The Cy5 wobbles before the assigned Cy5-S15 binding in the trace of Fig. 1d, which the reviewer has mentioned above, is due to protein stickiness because there are no corresponding anticorrelated changes in the Cy3 intensity. We discuss this trace in suppl. Fig. S11d and present also a metric which correctly assigns the bound state also in presence of stickiness. The lower Cy3 intensity of the Cy3-oligo after Cy5-S15 protein dissociation could be due to a local change in chemical environment of the Cy3 dye (protein induced fluorescence enhancement; Hwang & Myong, PNAS, 2011). We sometimes observe Cy3 intensity changes. For clarity for the reader, we have shifted the traces shown in Fig. 1d,e to supplementary Fig. S11, where we discuss how to assign states in traces with sticky proteins and have replaced them with two new traces performing background and bleed-through correction and now also show the FRET efficiency and a histogram of the FRET efficiency distribution in Fig. 1d,e.

In addition, to rule out that the Cy5 wobbles mentioned by the reviewer are not due to binding of S15 in a low-FRET or zero-FRET state, we have performed an additional

experiment with direct Cy5-S15 detection. This experiment confirms the absence of initial S15 binding in a low FRET state and is presented in supplementary Fig. S12a and we have also added an additional small section into the main text results section (underlined):

“We observe an immediate transition into a high-FRET state with apparent FRET efficiencies of ~0.5-0.6. This is a bit lower than expected from the distance between the Cy3 and Cy5 dyes in the native complex but is in agreement with a decreased FRET value observed in the ZMW compared to TIRF due to quenching of the Cy5 fluorescence by the ZMW aluminum walls (Chen & Puglisi, PNAS, 2014 and Uemura & Puglisi, Nature, 2010). The primary binding protein S4 has been shown to bind first in a low-FRET state before switching to the native high-FRET state within ~ 1 second (Kim & Woodson, Nature, 2014). There is no evidence for such an initial binding in a low-FRET state indirectly observing Cy5-S15 by FRET from the Cy3-oligo. To further address possible binding through an intermediate low-FRET state, we repeated our experiments using direct excitation of both Cy3 and Cy5 dyes at low 2 nM Cy5-S15 concentration (Supplementary Fig. 12a). We see a direct binding into the high-FRET state and no initially bound S15 protein without FRET in > 95 % of the traces. Overall, those experiments support the binding of S15 directly into a high-FRET state, corresponding to the native RNA conformation.”

We have added following new methods section (underlined):

“For the Cy5-S15 binding experiments, binding was detected by the presence of a strong FRET signal between Cy5-S15 and the Cy3-oligo binding to the 5'-end or 3'-end of the nascent RNA (Fig. 1 and Fig. 5b). We calculated the FRET efficiency by $E_{\text{FRET}} = I_A / (I_A + I_D)$ from background and spectral bleed-through corrected traces (Supplementary Fig. S9), where I_D and I_A are the apparent fluorescence intensities of the donor and acceptor, respectively. To assign the bound state, we defined the threshold in the middle of the two FRET states (Supplementary Fig. S9 and S10), with subsequent manual inspection for traces that showed substantial Cy5-protein stickiness and therefore a non-zero FRET efficiency baseline (Supplementary Fig. S11). Traces with substantial Cy5-S15 stickiness show a wavy Cy5 fluorescence baseline, resulting in a wavy FRET baseline (see Supplementary Fig. S11). For those traces, we manually had to adapt the FRET threshold to assign a bound state. We use the presence of a clear anti-correlated change in Cy3 and Cy5 intensities, which can be readily distinguished from Cy5-protein stickiness (see Supplementary Fig. 11a) due to the presence of a strong FRET in the bound state by design (see Fig. 5b). Alternatively, to assign anti-correlated Cy3-Cy5 intensities automatically, we use, instead of FRET, another metric defined by $I_A(t) * [I_D(t) - I_D(\text{max})]^2$, where $I_D(\text{max})$ is the intensity of the Cy3-oligo in absence of FRET. This metric allows us to assign automatically a bound state using a single threshold per trace even for traces with very strong Cy5-protein stickiness (see Supplementary. Fig. 11d-f).

7. The authors say they observe transient binding of S15 in the P class subset. In Figure 1e, how do the authors distinguish dissociation of S15 from fluctuations into a low FRET conformation? (Both scenarios would produce anti-correlated changes in Cy3 and Cy5 intensity; it can be hard to distinguish zero FRET from low FRET even with background correction. Maybe one could test whether P class S15 molecules exchange with free S15.)

This is an excellent question by the reviewer. That FRET fluctuations are rebinding events and not a conformational change during a long binding event are supported by the following observations:

First, the Cy3-oligo binding site has been designed such that there is a close 43 Å distance between Cy5-S15 and the Cy3-oligo. As mentioned above, S15 can only bind if the 3-way junction is formed and thus, S15 binding has to be observed as a strong FRET signal. We note that even if the 3-way junction is in the open form, this distance should not change a lot with our construct design. Thus, S15 binding should always be detected as a strong FRET.

Second, we see a concentration dependent increase in the binding rate for the P-type RNA molecules (see supplementary Fig. S6e,f), which is in agreement with rebinding events and not a conformational change, which would not show a concentration dependence.

Finally, to rule out entirely that the observed FRET fluctuations between high-FRET and zero-FRET are rebinding events, we have performed an additional experiment with direct Cy5-S15 detection. This experiment (described in Supplementary Fig. S12b and added as short section in main text) confirms rebinding events rather than a conformation change. The additional section added in the main text result section is (underlined):

“The Cy5-S15 binding rate for the P-class RNA molecules depends linearly on the Cy5-S15 concentration, with an apparent second order binding rate constant of $4.8 \pm 0.4 \times 10^6 \text{ M}^{-1} \text{ s}^{-1}$ (Supplementary Fig. S6e,f). This concentration dependence is in agreement with the fluctuations between zero-FRET and high-FRET corresponding to rebinding events rather than conformational changes during a long S15 binding event. To confirm the presence of multiple rebinding events, we have repeated the experiments with direct excitation of both Cy3 and Cy5 dyes at low 2 nM Cy5-S15 concentration (Supplementary Fig. S12b). No Cy5 fluorescence was detected between the short high-FRET state binding events, supporting the interpretation of multiple rebinding events.”

Additionally, the trace illustrating the P-class in Supp. Figure S5 is barely above the noise. The anti-correlated peaks indicate that FRET is occurring. Since the background is not shown, however, it is difficult to determine whether this is legitimate binding. The figure legend indicates “high FRET”, yet this cannot be said unless the traces are appropriately background corrected.

We have now corrected those traces for background and show the FRET efficiencies as an additional plot confirming the transitions between a low-FRET and high-FRET state for both N-type and P-type RNA molecules.

Minor points:

8. The labeling schemes outlined in Fig 2a-d are wonderful and clearly a lot of work has been done to characterize this system. However, the figure should be adjusted to be more easily understood (i.e. bigger traces) and there needs to be more discussion on each labeling scheme in the legend and in the text. For example, experiments in which both the DNA and the oligo are labeled with the same dye are initially confusing unless one carefully examines the cartoons for each panel.

We have modified Fig. 2a-d by making bigger traces and better illustrating the molecular events (dye movements during transcription elongation and binding of labeled DNA oligo). We have also corrected the traces for background and bleed-through as discussed in point 4 of reviewer #2. Finally, we have written a more detailed figure caption (underlined):

“Different DNA template labeling schemes result in different fluorescence intensity profiles. The DNA template labeling schemes are shown at the top of each trace with arrows showing the movement of the corresponding dyes towards or away from the surface during transcription progression resulting into a fluorescence intensity increase or decrease, respectively. The times of NTP injection and of labeled DNA oligo binding are indicated with arrows. The DNA oligo binds to the 3'-end of the nascent RNA to score for full-length RNA as illustrated in Fig. 1. Note that the characteristic fluorescence intensity changes also allow using a single color for monitoring transcription progression and scoring of full-length RNA with the labeled DNA oligo, i.e. Cy3 in (c,d).”

9. Please indicate the time when NTPs were injected in each trajectory; if this equals time 'zero', please state this somewhere. Also, is there a substantial delay between the addition of NTPs and the restart of transcription? I would expect transcription restarts in less than 1 s, but perhaps this is not the case.

We have now indicated in the trajectories or the figure captions when the NTPs have been delivered. Typically, transcription starts within a few seconds after NTP delivery, which is consistent with the mixing of the delivery mix. We have added an additional supplementary figure S1g with a histogram displaying the time from NTP injection till start of TK for a typical experiment and have added a sentence in the main text (changes underlined): “Consistent with a transcription elongation-dependent movement of the template 3'-end into the excitation volume close to the ZMW surface (Fig. 1c), we observed a monotonic increase in the Cy3.5 signal (Fig. 1d,e and Supplementary Fig. S8), typically starting within a few seconds after NTP delivery (Supplementary Fig. S1g).”

10. The 50 nt initial transcribed sequence used to generate the stalled elongation complexes and the long promoter region would allow for at least 2 RNAP to load onto the DNA template (based on the 23-25 bp footprint of *E. coli* RNAP). Could the authors note somewhere that this is a possibility and that it would complicate detection of protein binding during transcription?

The stalled complex is incubated with heparin after its formation, which results into the dissociation of all RNAP molecules bound to the DNA promoter region but not having successfully initiated transcription. Thus, the long promoter region does not increase the number of RNAPs that can be loaded on the DNA template. Only the number of RNAPs that can form a stable stalled transcription elongation complex until reaching the stalling site have to be considered.

However, using AFM it has been shown that the closest contacts between two RNAP molecules on the same DNA template is around 63 bp (Crampton & Thomson, NAR, 2006 or also see: Rivetti & Bustamante, JMB, 2003), which is not as close as one would expect from a hard sphere contact based separation of ~ 30-35 bp when inspecting the structure of an elongating RNAP. This measured separation would only allow the formation of a stalled transcription elongation complex containing a single RNAP per DNA template. This is in agreement with our native gels showing mainly a single band for the stalled transcription elongation complex (Suppl. Fig. S1b). We note that even if the nascent RNA in the stalled transcription elongation complex would be longer (e.g. > 100 nts) to allow the binding of two RNAPs per DNA template and thereby resulting in the transcription of two nascent RNA molecules from the same DNA template, those molecules would be sorted out during data evaluation: The presence of two nascent RNAs on the same DNA template would result in the binding of two Cy3-oligos to the 3'-end (or 5'-end) of the nascent RNAs and we have only evaluated traces showing a single Cy3-oligo binding event.

To clarify we have added in the methods section (changes underlined):

“A stalled transcription elongation complex, composed of a DNA template, an RNAP molecule and a nascent RNA of 50 nucleotides (Fig. 1a), was formed by incubation of 25 nM (labeled) DNA transcription.....Then, to prevent transcription reinitiation and for

disruption of unspecific RNAP/DNA interactions that could lead to the loading of more than one RNAP molecule per DNA template, 1 mg/ml heparin (or 20 µg/ml rifampicin) was added, followed by incubation for a further 20 minutes at 37° C. Using native gels, we verified the presence of a single RNAP molecule per DNA template in the stalled transcription-elongation complex (Supplementary Fig. 1b), which is in agreement with the closest contacts between two RNAP molecules on the same DNA template being around 63 bp (Crampton & Thomson, NAR, 2006)."

"...only traces showing a fluorescence intensity increase or decrease, which is characteristic for transcription elongation, and with the subsequent binding of a single Cy3-labeled DNA oligonucleotide, which is binding to the 3'-end of the nascent RNA and indicates the presence of a single full-length RNA, were selected for further analysis.

11. Suppl. Discussion 6. "...Cy3 dyes are predicted to clash with the RNA entry channel..." should be DNA entry channel.

This typo has been corrected.

12. I found the presentation of the S15 binding kinetics a bit confusing. The on-rate is slower than diffusion controlled; is this because binding is multi-step? Could multi-step binding explain why the observed on-rates do not increase linearly with S15 concentration? Or, do the authors think that the slower observed binding rate reflects a misfolded RNA population? It should be possible to account for the latter by separately analyzing N class and P class binding kinetics. Also, does Fig. S6f report the fast or slow observed binding rate?

We have separately analyzed the on-rate for the P-class and N-class RNA molecules. Figure S6f reports on the concentration dependence of the on-rate (fitted to a single-exponential function) for the P-class RNA molecules (between the multiple short binding events). This is stated both in the main text and in the corresponding figure caption:

"e,f, The measured on-rate between the multiple short binding events (s-1) in the P-class RNA molecules is protein concentration dependent. All the arrival times between the short Cy5-S15 binding events were fit to a single exponential function."

We obtain an on-rate between the multiple short binding events for the P-class RNA molecules of $\sim 5 \cdot 10^6 \text{ M}^{-1} \text{ s}^{-1}$.

For the N-class RNA molecules, most traces have only a single long-lived binding event. However, we have discussed the on-rate from end of transcription (or from Cy5-S15 protein delivery for a pre-folded RNA) till appearance of first stable binding event

and obtain a similar on-rate of $\sim 6 \cdot 10^6 \text{ M}^{-1} \text{ s}^{-1}$ for the fast component of the double exponential fit (Suppl. Fig. S7a,b and suppl. Discussion S7). As discussed in Suppl. Discussion S7, our on-rates are in a similar range, though even slightly higher as in previous EMSA and other ensemble experiments under different conditions (references are provided in Suppl. Discussion S7).

The on-rate is likely slower than diffusion controlled because S15 protein binding requires the closure of the 3-way junction as stated and referenced in the main text and further supported in supplementary Fig. S5 showing that S15 leads to closure of the 3-way junction. This is the case for both N-type and P-type RNA molecules. Thus, binding is multistep and therefore can explain the slower than diffusion controlled on-rate.

We thank the reviewer for bringing this to our attention and have therefore added an additional sentence into the main text of the results section (underlined):

“Those data demonstrate that the RNA transitions into a native docked conformation, both during the stable S15 binding events in the N-class molecules (Supplementary Fig. 5c), and during the transient binding events in the P-class molecules (Supplementary Fig. 5d). This multistep binding mechanism is consistent with an on-rate for S15 binding that is slower than diffusion controlled, in both N-type (Supplementary Fig. 7a,b) and P-type (Supplementary Fig. 6e,f) RNA molecules.”

13. In Supp Figure S7, there is a “FRET” signal in Cy5 before the Cy3 oligo has bound to the RNA and before the DNA template has dissociated. Is this bleed through from the Cy3.5 channel? The signal appears to increase without a mirrored increase in Cy3.5, so why would there be an increase in the Cy5 signal unless this was an S15 binding event? The authors assign S15 binding concomitant with Cy3 signal but there is not a significant increase in the fluorescence signal in Cy5, in fact it appears to slowly increase over the entire movie. There does not seem to be any anti-correlation until possibly 160 s, near the end of the trace. Is this typical?

We now show the trace as background and bleed-through corrected trace and have moved this panel (Fig. S7c) to the new supplementary Fig. S11e where we are discussing how to assign states with protein stickiness (see also point 5 reviewer #2). The apparent increase of the Cy5 signal over time was on one hand due to the bleed-through from the Cy3.5 channel (~40-60sec) which is now corrected in the new figure. On the other hand, we sometimes see an increase of the Cy5 signal over time for high concentrations of sticky proteins. After background correction this trace may show a slight increase in Cy5 signal over time (~70-100sec) but this may also be a slight baseline drift.

14. Also in Figure S7c, what is the meaning of the three panels on the right? They don't seem to be connected with any other panel.

This panel was intended to support the two traces shown on the left. As they are confusing and do not add additional information, we have removed them from the figure.

REVIEWERS' COMMENTS:

Reviewer #1 (Remarks to the Author):

The authors made a commendable effort revising the manuscript and comprehensively addressed concerns raised by the reviewers. However, I still think that the state assignments in single-molecule traces ideally should be done using unbiased analysis tools instead of a manual threshold analysis. Nevertheless, the manuscript can be accepted for publication in its current form with minor revisions suggested below.

I recommend replacing the term “strong FRET signal” used by the authors throughout the manuscript with more appropriate phrasing “high FRET value”. The authors should also specify which FRET value they consider to be the “strong FRET signal”. The authors should also replace the term “stickiness” with more appropriate term “non-specific binding”. They need to clarify how non-specific binding was distinguished from specific binding. Was the presence of Cy5 fluorescence in the absence of reciprocal decrease in Cy3 fluorescence used as a signature of non-specific S15-Cy5 binding?

Reviewer #2 (Remarks to the Author):

Overall, the authors have addressed the reviewer’s comments by better explaining the method in the main text, by adding text and figures to the supplement that detail the data analysis, and by correcting their traces for fluorescence background and bleed-through. There are some inherent limitations to the method, such as the need to hybridize an oligomer to the RNA at an appropriate position that can distinguish specific and non-specific S15 protein binding (although the experiment in Fig. 6 is really nice). The method can only accommodate a limited window of transcript lengths. Nevertheless, the method provides an exciting new way to investigate transcription-coupled assembly of RNA-protein complexes, and its pros and cons should be more apparent to readers with these revisions. I think this study will certainly interest a wide spectrum of readers.

I recommend just a few remaining changes:

1. The authors have included additional text describing their method for determining the beginning and end of transcription, which is great. As there is some uncertainty in the manual assignment method, they should clearly state this in the main text and provide a window for the transcription speed based on this uncertainty (17-25 nt/s).
2. For clarity, the authors should state in the figure legends which lasers are on during each experiment – if they excite only Cy3 and Cy3.5 in all experiments except for Fig. S12, then they should state this in Fig. 1, at least, and again in Fig. S12. (Obviously, direct excitation of Cy5 affects FRET measurements.)
3. Fig 6 describing experiments with the 5 Cy3-oligo does not show traces for P class molecules; is this because they don’t observe transient S15 binding in this experiment or because they just happened not to include an example of it in the figure? If transient binding is a frequent behavior, as implied by the cartoon in Fig. 7 and the experiment in Fig. S12 (all those red peaks), it would be best to include an example in Fig. 6 -- for completeness, and to bolster the claim that the location of the oligo binding site does not affect S15 binding specificity.
4. As an aside, Fig. S12b nicely illustrates why accurate background subtraction is so crucial to the

interpretation of the S15 binding events (zero FRET vs. low FRET). (Not quite sure how they did this with the red laser on.)

Reviewer #1:

The authors made a commendable effort revising the manuscript and comprehensively addressed concerns raised by the reviewers. However, I still think that the state assignments in single-molecule traces ideally should be done using unbiased analysis tools instead of a manual threshold analysis. Nevertheless, the manuscript can be accepted for publication in its current form with minor revisions suggested below.

I recommend replacing the term “strong FRET signal” used by the authors throughout the manuscript with more appropriate phrasing “high FRET value”. The authors should also specify which FRET value they consider to be the “strong FRET signal”. The authors should also replace the term “stickiness” with more appropriate term “non-specific binding”. They need to clarify how non-specific binding was distinguished from specific binding. Was the presence of Cy5 fluorescence in the absence of reciprocal decrease in Cy3 fluorescence used as a signature of non-specific S15-Cy5 binding?

We have replaced “strong FRET signal” by “high-FRET value” and replaced the term “stickiness” by “non-specific binding” throughout the main text including supplementary information.

We have specified in the main text what we consider as high-FRET value: “We observe an immediate transition into a high-FRET state with apparent FRET efficiencies of ~0.5-0.6.”

We have specified in the methods section how we distinguish non-specific binding from specific binding: “We use the presence of a clear anti-correlated change in Cy3 and Cy5 intensities, which can be readily distinguished from non-specific Cy5-protein binding..... In contrast, non-specific Cy5-protein binding results only in a Cy5 fluorescence increase in absence of a reciprocal decrease in Cy3 fluorescence.”

Reviewer #2:

Overall, the authors have addressed the reviewer’s comments by better explaining the method in the main text, by adding text and figures to the supplement that detail the data analysis, and by correcting their traces for fluorescence background and bleed-through. There are some inherent limitations to the method, such as the need to hybridize an oligomer to the RNA at an appropriate position that can distinguish specific and non-specific S15 protein binding (although the experiment in Fig. 6 is really nice). The method can only accommodate a limited window of transcript lengths. Nevertheless, the method provides an exciting new way to investigate transcription-coupled assembly of RNA-protein complexes, and its pros and cons should be more apparent to readers with these revisions. I think this study will certainly interest a wide

spectrum of readers.

I recommend just a few remaining changes:

1. The authors have included additional text describing their method for determining the beginning and end of transcription, which is great. As there is some uncertainty in the manual assignment method, they should clearly state this in the main text and provide a window for the transcription speed based on this uncertainty (17-25 nt/s).

We have added following text to the main text methods:

“Depending on the signal to noise ratio of the trace and the length of the RNA construct, the determination of the exact moment for start and end of transcription can have an uncertainty of up to a 2-3 seconds. This translates to an uncertainty for the determination of the transcription rate of single RNA molecules and is more pronounced for smaller RNA molecules (< 200 nt). As an example, a specific RNA molecule of 200 nt with a transcription time of 10 seconds (Fig. 3b) has an average transcription rate of 20 nt s⁻¹. If the determination of the transcription time of this specific RNA molecule has an uncertainty of ± 2.5 seconds due to a low signal to noise ratio of that trace, this results in a range for the average transcription rate of 16-26.7 nt s⁻¹ for this specific RNA molecule. We note that with the large amount of data generated by the ZMW technology it is possible to select only traces with a high signal to noise ratio, thereby reducing the uncertainty in the determination of the transcription rate at the single-molecule level.”

2. For clarity, the authors should state in the figure legends which lasers are on during each experiment – if they excite only Cy3 and Cy3.5 in all experiments except for Fig. S12, then they should state this in Fig. 1, at least, and again in Fig. S12. (Obviously, direct excitation of Cy5 affects FRET measurements.)

We have added to the figure legend of Fig.1: “A single laser at 532 nm was used to directly excite both the Cy3 and Cy3.5 dyes.”

We have added to the figure legend of Fig.2: “532 nm and 642 nm lasers were used to excite the Cy3 and Cy5 dyes, respectively (a,c,d). A single laser at 532 nm was used to excite both the Cy3 and Cy3.5 dyes (b).”

We have added to the figure legend of Fig.4: “532 nm and 642 nm lasers were used to excite the Cy3 (green) and Cy5 (red) dyes, respectively.”

We have added to the figure legend of Fig.6: “A single laser at 532 nm was used to directly excite both Cy3 and Cy3.5 dyes.”

We have added to the figure legend of Suppl. Fig.5: “A single laser at 532 nm was used to excite the Cy3 dye.”

We have added to the figure legend of Suppl. Fig.12: “Therefore, both 532 nm and 642 nm lasers were used to excite the Cy3 and Cy5 dyes, respectively.”

3. Fig 6 describing experiments with the 5 Cy3-oligo does not show traces for P class molecules; is this because they don't observe transient S15 binding in this experiment or because they just happened not to include an example of it in the figure? If transient binding is a frequent behavior, as implied by the cartoon in Fig. 7 and the experiment in Fig. S12 (all those red peaks), it would be best to include an example in Fig. 6 -- for completeness, and to bolster the claim that the location of the oligo binding site does not affect S15 binding specificity.

For completeness, we have made an additional supplementary Fig. 13, showing representative P-class RNA molecule traces occurring also when the Cy3-oligo is hybridized to the 5'-end of the nascent RNA to detect specific Cy5-S15 binding.

4. As an aside, Fig. S12b nicely illustrates why accurate background subtraction is so crucial to the interpretation of the S15 binding events (zero FRET vs. low FRET). (Not quite sure how they did this with the red laser on.)

We are happy that the reviewer values our additional data. The Cy5 background in presence of the red laser can be obtained from the intensity before the injection of the delivery mix containing NTPs, the Cy3 oligo and the Cy5-S15 protein.